# FUNCTIONAL DISTRIBUTION NETWORKS (FDN)

## ABSTRACT

Modern probabilistic regressors often remain overconfident under distribution shift. Functional Distribution Networks (FDN) place input-conditioned distributions over network weights, producing predictive mixtures whose dispersion adapts to the input; we train them with a Monte Carlo $\beta$–ELBO objective. We pair FDN with an evaluation protocol that separates interpolation from extrapolation and emphasizes simple OOD sanity checks. On controlled 1D tasks and small/medium UCI-style regression benchmarks, FDN remains competitive in accuracy with strong Bayesian, ensemble, dropout, and hypernetwork baselines, while providing strongly input-dependent, shift-aware uncertainty and competitive calibration under matched parameter and update budgets.

## 1 INTRODUCTION

Modern neural predictors are routinely deployed under dataset shift, where test inputs depart from the training distribution. In these regimes, point predictions from deterministic networks and naïve uncertainty surrogates from traditional stochastic heuristics often become *overconfident*—assigning high probability to wrong outcomes, particularly off-support/extrapolation—undermining reliable decision making. Bayesian Neural Networks (BNNs), MLP with dropout, Deep Ensembles, and Hypernetworks are strong practical baselines, yet they can still under-react outside the training support or require substantial ensembling/sampling to behave robustly (Quiñonero-Candela et al., 2009; Ovadia et al., 2019; Guo et al., 2017; MacKay, 1992; Neal, 1996; Graves, 2011; Blundell et al., 2015; Gal & Ghahramani, 2016; Lakshminarayanan et al., 2017; Ha et al., 2017).

This motivates architectures that are *uncertainty-aware* and *calibrated*—sharp in-distribution (ID) and widening appropriately OOD (higher CRPS, wider intervals). We pursue this with *Functional Distribution Networks (FDN)*, which place input-conditional distributions over weights to modulate uncertainty locally in $x$. Concretely, FDN amortizes an input-conditional posterior $q_\phi(\theta \mid x)$ via small Hypernetworks and trains it with a Monte Carlo likelihood and a $\beta$-ELBO. Under matched budgets, FDN is ID-competitive and well-calibrated by *scale* on controlled 1D shifts: on low-frequency or piecewise-smooth tasks (step, quadratic) it achieves near-unity MSE–Var slopes with high Spearman correlation and favorable AURC/CRPS, indicating that predictive variance tracks error growth both in- and out-of-distribution. On more oscillatory shifts (sine) FDN preserves strong rank calibration and increases variance OOD, but its variance still under-scales relative to the error—an explicit target for future improvement. Beyond 1D toy functions, we evaluate We further evaluate FDN on standard small/medium UCI-style regression benchmarks—Airfoil Self-Noise, Combined Cycle Power Plant, and Energy Efficiency (Brooks et al., 1989; Tüfekci & Kaya, 2014; Tsanas & Xifara, 2012)—under feature-based ID/OOD splits. On these real datasets FDN remains ID-competitive and typically exhibits large positive $\Delta$Var with moderate $\Delta$MSE and $\Delta$CRPS, i.e., it widens uncertainty under shift while maintaining sensible accuracy and calibration relative to strong Bayesian, ensemble, and hypernetwork baselines.

**Overview.** Rather than treating weights as fixed (or globally random), FDN places an input-conditioned distribution over weights:

$$\theta \mid x \sim p(\theta \mid x), \qquad y \mid x, \theta \sim p(y \mid x, \theta).$$

For tractability we fix an input-agnostic prior $p(\theta \mid x) = p_0(\theta) = \mathcal{N}(0, \sigma_0^2 I)$, so all input dependence is carried by an amortized posterior $q_\phi(\theta \mid x)$ implemented via small Hypernetworks. Many standard uncertainty-aware architectures can be cast in this template via different choices of

$q_\phi(\theta \mid x)$; we detail these instantiations in Appendix A. Sampling $\theta \sim q_\phi(\theta \mid x)$ yields locally adaptive functions, and as $x$ moves off the training support the induced weight distribution can broaden, producing wider and more appropriately uncertain predictive densities.

**Training and evaluation.** We train with a Monte Carlo $\beta$–ELBO over $q_\phi(\theta \mid x)$ and also report an IWAE variant; explicit expressions for the layer-wise KL and its LP-FDN factorization are given in the Appendix B. Our evaluation protocol splits test points into interpolation (ID) and extrapolation (OOD) regions and summarizes shift via deltas $\Delta(\cdot) = \mathbb{E}_{\text{OOD}}[\cdot] - \mathbb{E}_{\text{ID}}[\cdot]$, focusing on $\Delta\text{MSE}$, $\Delta\text{Var}$, and $\Delta\text{CRPS}$ together with MSE–variance fits and risk–coverage curves.

**Contributions.** (i) **Model.** We introduce Functional Distribution Networks (FDN), a simple module that amortizes input-conditioned weight distributions via small Hypernetworks, in two variants: IC-FDNet (conditioning on $x$) and LP-FDNet (conditioning layer-wise on previous activations). (ii) **Protocol.** We propose a small-suite extrapolation protocol that targets $\Delta\text{Var} > 0$ under shift and complements it with calibration diagnostics (MSE–variance slope, rank correlation, and risk–coverage). (iii) **Empirical study.** Under matched parameter, update, and predictive-sample budgets, we benchmark FDN against strong Bayesian, ensemble, dropout, and hypernetwork baselines on controlled 1D function families and UCI-style regression tasks, showing that FDN remains ID-competitive while providing practically useful, shift-aware uncertainty.

**Scope.** We focus on low-dimensional regression with homoscedastic scalar Gaussian heads and relatively shallow backbones in order to isolate ID vs. OOD behavior under tightly matched budgets. All code, configurations, and scripts to reproduce the experiments will be released upon acceptance.

## 2 RELATED WORK

**Uncertainty in neural regression.** BNN place distributions over weights and infer posteriors via variational approximations or MCMC (MacKay, 1992; Neal, 1996; Graves, 2011; Blundell et al., 2015). Deep Ensembles average predictions from independently trained networks and are a strong practical baseline (Lakshminarayanan et al., 2017; Maddox et al., 2019). MLP (with dropout) interprets dropout at test time as approximate Bayesian inference (Gal & Ghahramani, 2016). Heteroscedastic regression learns input-dependent output variance but retains deterministic weights (Nix & Weigend, 1994; Kendall & Gal, 2017).

**Hypernetworks and conditional weight generation.** Hypernetworks generate the weights of a primary network using an auxiliary network (Ha et al., 2017); related work explores dynamic, input-conditioned filters and conditional computation (De Brabandere et al., 2016; Brock et al., 2018). Bayesian/uncertainty-aware Hypernetworks place distributions over generated weights and train them variationally (Krueger et al., 2017). Our FDN differs by explicitly conditioning the weight distributions on the current input (or intermediate activations) in order to modulate epistemic uncertainty itself, not only deterministic weights. This input-aware weight stochasticity is what enables FDN to widen uncertainty under distributional shift while maintaining competitive ID accuracy under matched budgets.

**Meta-learning and context-conditioned predictors.** Neural Processes (NP) learn a distribution over functions conditioned on a context set $\mathcal{C}$ via a global latent, yielding $p(y \mid x, \mathcal{C})$ (Garnelo et al., 2018; Kim et al., 2019); gradient-based meta-learning (e.g., MAML) instead adapts an initialization per task via inner-loop gradients (Finn et al., 2017). FDN is complementary: it amortizes *local* weight uncertainty directly on $x$ (and optionally a compact context) through $q_\phi(\theta \mid x, c)$, requires no inner-loop adaptation, and injects uncertainty at the layer level rather than via a single global latent.

**Calibration and OOD behavior.** Proper scoring rules such as the continuous ranked probability score (CRPS) are strictly proper and reward calibrated predictive distributions (Gneiting & Raftery, 2007). Empirical studies highlight overconfidence under dataset shift and introduce OOD benchmarks (Ovadia et al., 2019). Our evaluation therefore separates interpolation from extrapolation and uses the monotonic relationship between per-sample squared error (MSE) and predicted variance as a simple diagnostic calibration check.

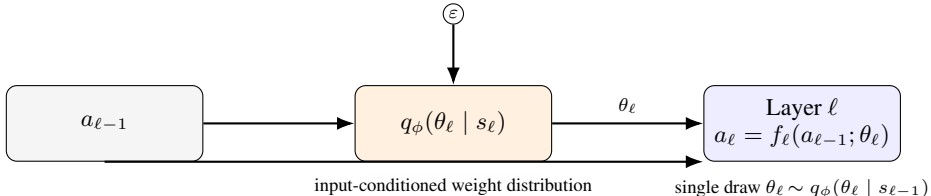

Figure 1: Single-layer view of FDNet. For layer $\ell$, the previous activation $a_{\ell-1}$ (with $a_0 = x$ for the first layer) is fed to both the hypernetwork and the main layer. For IC-FDNet $s_\ell = x$ while for LP-FDNet $s_\ell = a_{\ell-1}$. The hypernetwork takes $s_\ell$ and a random draw $\varepsilon$ to generate weights/ biases (Gaussian Head with reparametrization technique) $\theta_\ell = (W_\ell, b_\ell)$, which are then used by the main layer $a_\ell = f_\ell(a_{\ell-1}; \theta_\ell)$.

**Positioning.** Compared to BNNs, FDN sidesteps global posteriors by amortizing *local* weight distributions $q_\phi(\theta \mid x)$. Compared to Deep Ensembles, FDN uses shared parameters and stochastic generation instead of replicating full models. Compared to Hypernetworks, FDN explicitly models *uncertainty over* generated weights and regularizes it with a KL prior, enabling principled OOD expansion.

## 3 METHOD

For convenience, a summary of the main symbols and dimensions used throughout the paper is provided in Table 1 in Appendix C.

### 3.1 PRELIMINARIES

We model $y \in \mathbb{R}^{d_y}$ with a neural network $f_\theta : \mathbb{R}^{d_x} \to \mathbb{R}^{D_y}$ ($D_y = 2d_y$ for a diagonal covariance and $D_y = d_y(d_y + 1)/2$ for full covariance) whose final layer parameterizes a Gaussian predictive head,

$$f_\theta(x) = \big(\mu_\theta(x), \Sigma_\theta(x)\big), \qquad p(y \mid x, \theta) = \mathcal{N}\big(y; \mu_\theta(x), \Sigma_\theta(x)\big),$$

In the main experiments we restrict to scalar outputs and use a homoscedastic Gaussian likelihood $p(y \mid x, \theta) = \mathcal{N}\big(y; \mu_\theta(x), \sigma^2\big)$ with fixed variance $\sigma^2$, shared across all models. For such a head the per-example contribution to the $\beta$–ELBO reduces to a rescaled squared-error term plus the KL penalty between the input-conditioned weight posterior and the prior; constants can be absorbed into the trade-off parameter $\beta$. We therefore optimize a weighted $MSE + \beta KL$ objective and use the same Gaussian head for all baselines to ensure a fair comparison. More general multivariate, heteroscedastic, and structured-covariance likelihoods are discussed in Appendix D.

### 3.2 FDN: INPUT-CONDITIONED WEIGHT DISTRIBUTIONS

We drop explicit context and condition only on signals from the network itself. For each layer $\ell$, FDN places a diagonal-Gaussian over its weights whose parameters are produced by a small Hypernetwork $A_\ell(\cdot)$. We choose the conditioning signal

$$s_\ell^{(k)} \in \begin{cases} x, & \text{IC-FDN} \\ a_{\ell-1}^{(k)}, & \text{LP-FDN (with } a_0^{(k)} = x), \end{cases}$$

and set

$$(\mu_{W,\ell}, \rho_{W,\ell}, \mu_{b,\ell}, \rho_{b,\ell}) = A_\ell(s_\ell^{(k)}), \qquad \sigma_{W,\ell} = \varepsilon + \text{softplus}(\rho_{W,\ell}), \quad \sigma_{b,\ell} = \varepsilon + \text{softplus}(\rho_{b,\ell}),$$

with a small floor $\varepsilon = 10^{-3}$ for numerical stability. Sampling then proceeds as

$$W_\ell^{(k)} = \mu_{W,\ell} + \sigma_{W,\ell} \odot z_{W,\ell}^{(k)}, \qquad b_\ell^{(k)} = \mu_{b,\ell} + \sigma_{b,\ell} \odot z_{b,\ell}^{(k)}, \qquad z_{\{\cdot\},\ell}^{(k)} \sim \mathcal{N}(0, I),$$

with the conditioning signal chosen as $s_\ell = x$ for IC-FDNet and $s_\ell = a_{\ell-1}^{(k)}$ for LP-FDNet, where in both cases the layer output is $a_\ell^{(k)} = f_\ell(a_{\ell-1}^{(k)}; W_\ell^{(k)}, b_\ell^{(k)})$ (sequential across layers). Figure 1

illustrates a single FDN layer: a small hypernetwork maps $s_\ell$ to the parameters of a Gaussian distribution over weights and biases; a sample $\theta_\ell = (W_\ell, b_\ell)$ from this distribution is then used to compute $a_\ell = f_\ell(a_{\ell-1}; \theta_\ell)$.

**Variational family (compact).** FDN uses a layer-wise diagonal-Gaussian over weights, conditioned on $s_\ell \in \{x, a_{\ell-1}^{(k)}\}$:

$$q_\phi(\theta \mid x) = \prod_{\ell=1}^{L} \mathcal{N}\big(\text{vec}(W_\ell); \mu_{W,\ell}(s_\ell), \text{diag}\, \sigma_{W,\ell}^2(s_\ell)\big)\, \mathcal{N}\big(b_\ell; \mu_{b,\ell}(s_\ell), \text{diag}\, \sigma_{b,\ell}^2(s_\ell)\big),$$

In compact form (concatenating all layers),

$$q_\phi(\theta \mid x) = \mathcal{N}\big(\theta; \mu_\phi(x), \text{diag}\, \sigma_\phi^2(x)\big), \qquad \theta^{(k)} = \mu_\phi(x) + \sigma_\phi(x) \odot \varepsilon^{(k)}, \ \ \varepsilon^{(k)} \sim \mathcal{N}(0, I),$$

and the predictive density is the Monte Carlo mixture

$$p(y \mid x) \approx \frac{1}{K} \sum_{k=1}^{K} p\big(y \mid x, \theta^{(k)}\big), \qquad \theta^{(k)} \sim q_\phi(\theta \mid x).$$

**Prior and regularization.** We regularize $q_\phi(\theta \mid x)$ toward a simple reference $p_0(\theta) = \prod_\ell \mathcal{N}(0, \sigma_0^2 I)$ via a $\beta$-weighted KL term (we use $\sigma_0 = 1$ in all experiments). For diagonal Gaussians,

$$D_{\text{KL}}\big(\mathcal{N}(\mu, \text{diag}\, \sigma^2) \,\big\|\, \mathcal{N}(0, \sigma_0^2 I)\big) = \tfrac{1}{2} \sum_j \left( \frac{\sigma_j^2 + \mu_j^2}{\sigma_0^2} \ - \ 1 \ - \ \log \frac{\sigma_j^2}{\sigma_0^2} \right),$$

and we sum this over all layers (for both $W_\ell$ and $b_\ell$). The variance floor is implemented by the $\varepsilon$ in $\sigma = \varepsilon + \text{softplus}(\rho)$ rather than a hard bound.

Algorithm 1 (Appendix E) summarizes training ($\beta$-ELBO with re-parameterized gradients) and inference (Monte-Carlo mixtures over weight draws) for both IC-FDN and LP-FDN.

**IC-FDNet vs. LP-FDNet.** We study two conditioning schemes. *IC-FDNet* (Input-Conditioned) uses the raw input $x$ at every layer, $s_\ell(x) = x$, so all layers see the same features when sampling weights. *LP-FDNet* (Layer-Progressive) instead uses hidden activations, with $s_0(x) = x$ and $s_\ell(x) = a_{\ell-1}(x)$ for $\ell \geq 1$, yielding a depth-aware conditioning scheme. Empirically, the two variants achieve similar in-distribution accuracy across our benchmarks, while LP-FDNet often produces somewhat larger increases in predictive variance under distribution shift (larger $\Delta\text{Var}$) at comparable MSE.

## 4 EXPERIMENTS

**Tasks and splits.** We evaluate FDN on (i) controlled 1D regression tasks where the ground-truth function is known and interpolation vs. extrapolation is precisely defined in input space, and (ii) standard small/medium UCI-style regression benchmarks with feature-based ID/OOD splits. In both settings we report metrics separately on the interpolation (ID) region, the extrapolation (OOD) region, and the aggregated test set, and we summarize distribution shift by deltas $\Delta(\cdot) = \mathbb{E}_{\text{OOD}}[\cdot] - \mathbb{E}_{\text{ID}}[\cdot]$.

### 4.1 TOY FUNCTION FAMILIES AND ID/OOD PROTOCOL

We first benchmark on 1D toy regression tasks, where the ground-truth function $f : \mathbb{R} \to \mathbb{R}$ is known. For each run we select one of three families:

- **Step**: $f(x) = H(x) = \mathbf{1}\{x \geq x_0\}$, with a discontinuity at $x_0 = 0$.
- **Sine**: $f(x) = A\sin(\omega x)$, with amplitude $A$ and frequency $\omega$.
- **Quadratic**: $f(x) = ax^2 + b$, with $(a, b)$ fixed across runs.

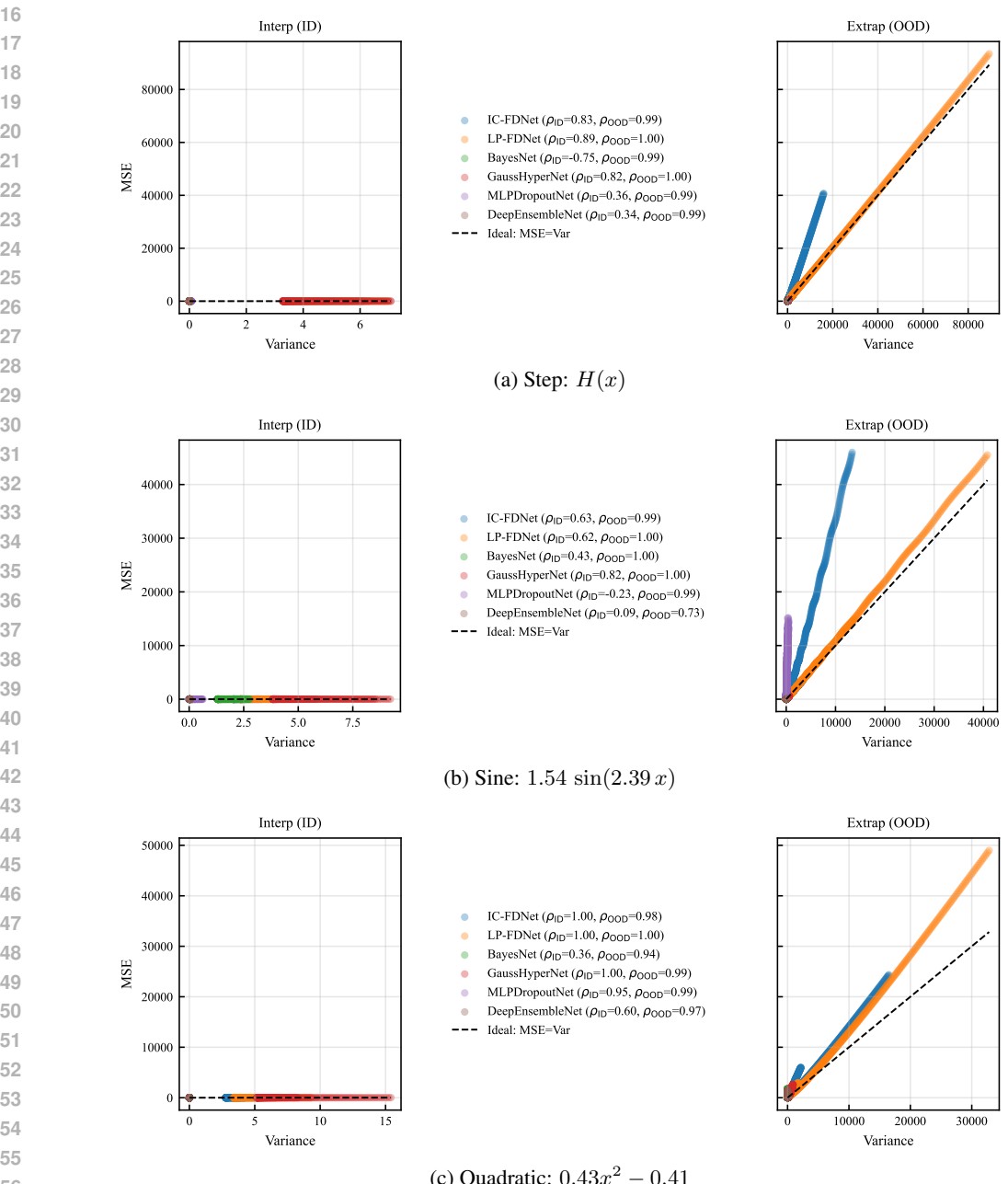

Figure 2: MSE vs. predicted variance on three 1D regression tasks (rows: step, sine, quadratic). Left/right panels in each row show interpolation (ID) and extrapolation (OOD) test points with shared axes; the dashed line marks the ideal $\text{MSE} = \text{Var}$. Legends report per-model Spearman's $\rho$ in each region.

Unless otherwise stated we use the specific instantiations shown in Figure 2 (step $H(x)$, sine $1.54 \sin(2.39x)$, and quadratic $0.43x^2 - 0.41$) to match all reported plots. We define a symmetric interpolation region $R_{\text{interp}} = [-\ell, \ell]$ and an extrapolation region $R_{\text{extrap}} = \mathbb{R} \setminus R_{\text{interp}}$. Training and validation inputs are sampled uniformly from $R_{\text{interp}}$; test inputs cover both $R_{\text{interp}}$ and $R_{\text{extrap}}$ on a dense grid. We report all metrics separately on the ID split (test points in $R_{\text{interp}}$), the OOD split (test points outside $R_{\text{interp}}$), and on the full test set. This protocol ensures that "OOD" strictly corresponds to extrapolation in input space.

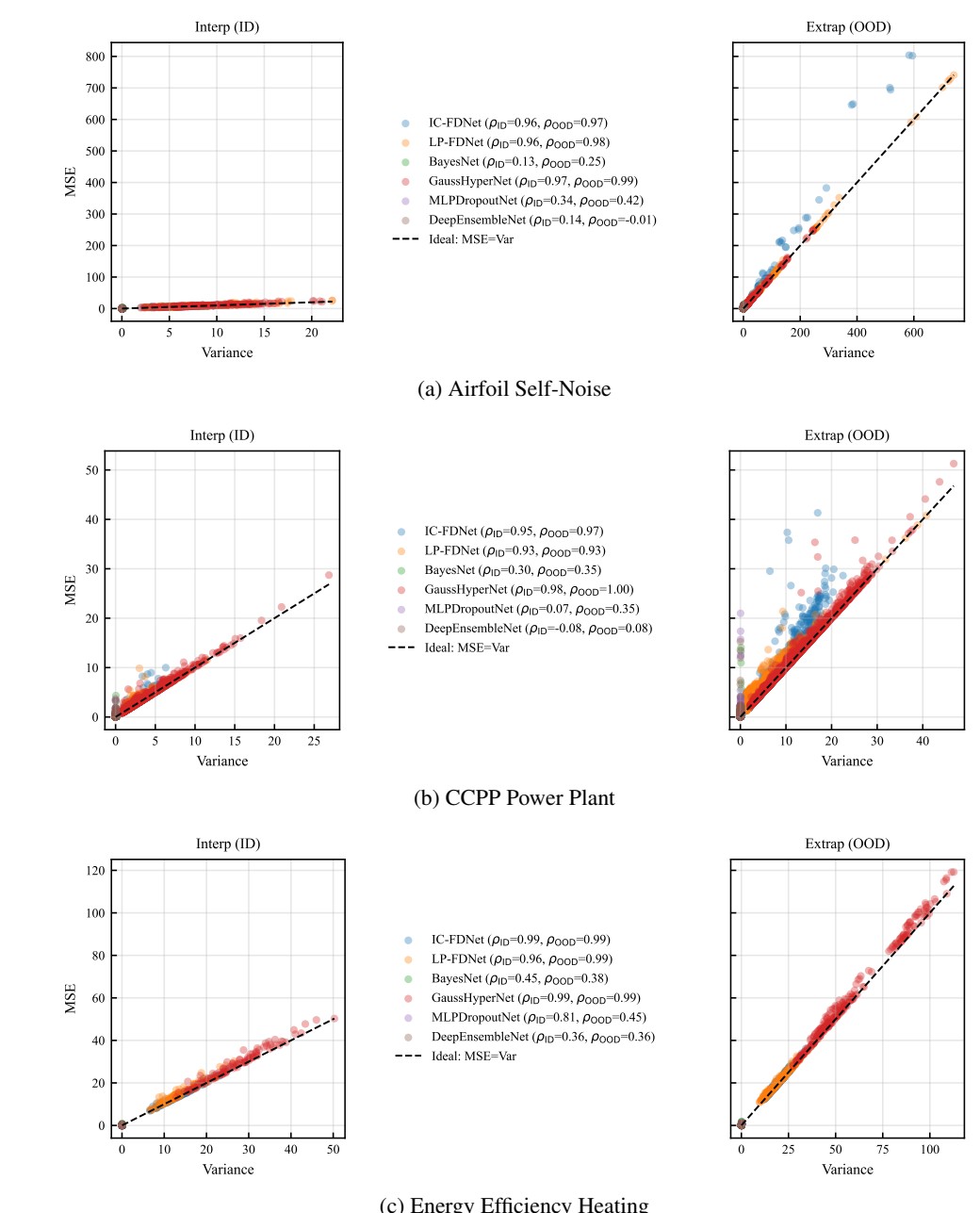

Figure 3: MSE vs. predicted variance on three real regression datasets: Airfoil Self-Noise, CCPP Power Plant, and Energy Efficiency (heating). For each dataset, left/right panels show ID and OOD test points with shared axes, and the dashed line marks $\mathrm{MSE} = \mathrm{Var}$. Legends report per-model Spearman's $\rho$.

## 4.2 REAL REGRESSION DATASETS AND ID/OOD SPLITS

To test FDNs beyond 1D toy functions we use three standard UCI regression datasets: Airfoil Self-Noise, Combined Cycle Power Plant (CCPP), and Energy Efficiency (heating load as the primary target). For each dataset we follow a consistent preprocessing and ID/OOD protocol: we choose a single "ID feature" $z$ with a natural interpretation (e.g., frequency for Airfoil, ambient temperature for CCPP, relative compactness for Energy), define the interpolation band as the 20th–80th percentiles of this feature and treat the extremes as extrapolation, split train/validation/test by quantiles of $z$, and

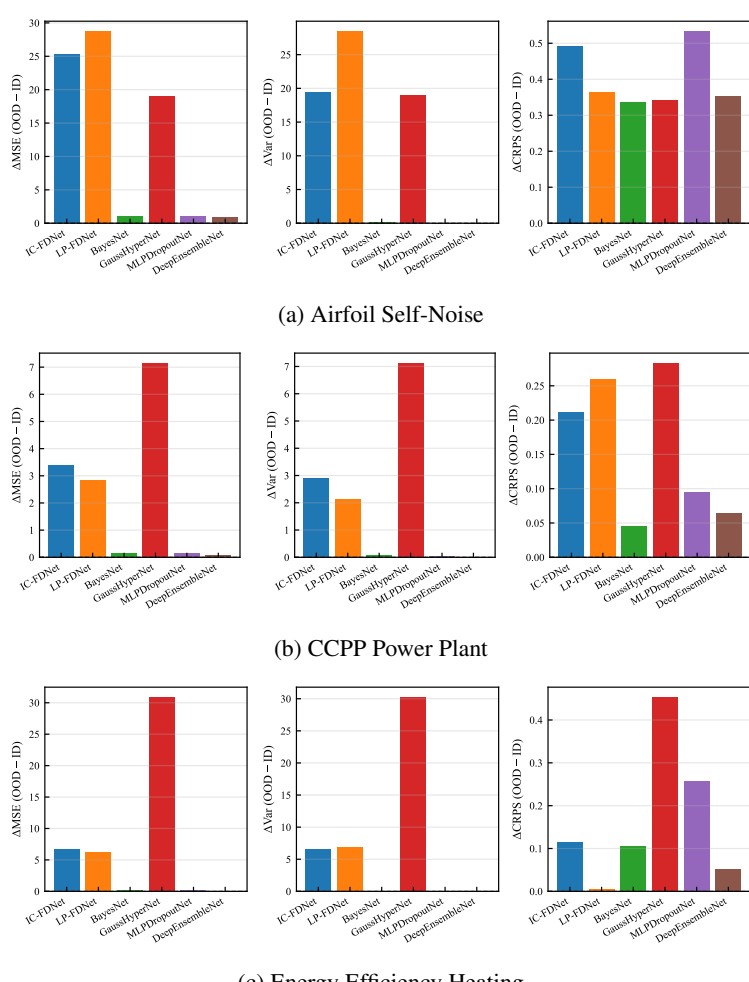

(a) Airfoil Self-Noise

(b) CCPP Power Plant

(c) Energy Efficiency Heating

Figure 4: ID→OOD deltas on the real regression datasets. Bars show $\Delta$MSE, $\Delta$Var, and $\Delta$CRPS for Airfoil, CCPP, and Energy. Well-behaved models exhibit large positive $\Delta$Var together with moderate $\Delta$MSE and $\Delta$CRPS, indicating that uncertainty widens under shift while accuracy and calibration degrade gracefully.

standardize all inputs and targets using training statistics. We fix a single train/validation/test split shared across all methods and seeds. Exact dataset sizes are summarized in Table 3 (Appendix F).

### 4.3 COMPLEXITY, CAPACITY, AND FAIRNESS

**Baselines.** We evaluate four stochastic baselines: MLP with dropout (**MLPDropoutNet**), Deep Ensemble of MLP (**DeepEnsemblesNet**), Variational BNN (**BayesNet**), and Gaussian Hyper-network (**GaussianHyperNet**). Because our study centers on calibrated *predictive distributions* (CRPS, MSE–Var slope/intercept, AURC), we omit the **Deterministic MLP** and the (input-conditioned) **Hypernetwork** from the main uncertainty analysis; their ID/OOD MSE is comparable to Ensembles/Dropout. Training details (optimizer, batch size, learning rate, prior scale $\sigma_0$, variance floor $\sigma_{\min}$) appear in Table 2 (Appendix F).

**Link-budget.** We consider networks with a single hidden layer and fix the parameter budget to $P \approx 1000$ ($\pm 5\%$) for all models, counting *all* trainable parameters, including any Hypernetwork components; counts appear in Table 4 (Appendix F). To equalize the *update* budget, ensembles with $M$ members use *epoch-split* training (epochs divided by $M$). For non-ensemble networks we use

one Monte Carlo draw per update ($K = 1$), keeping per-step cost comparable and the total number of parameter updates matched across models.

*Note:* To hit the $P$ target, MLPDropoutNet uses widened hidden layers, increasing capacity/expressive power and potentially improving ID MSE independent of uncertainty quality; hence our emphasis on calibration-centric metrics.

**Computational complexity and scalability.** For all models, the per-epoch training cost scales linearly in the number of examples $N$ and quadratically in the layer widths, $O(N \sum_\ell d_{\ell-1} d_\ell)$. Both IC-FDN and LP-FDN inherit this scaling from the base MLP: the per-layer Hypernetworks add only a small constant-factor overhead because their hidden width $h_{\text{hyp}}$ and the Monte Carlo count $K$ are fixed (parameter count remains $O(d_{\ell-1} d_\ell)$ per layer), matching the baselines. A full discussion of parameter-count and architectural trade-offs for larger models are given in Appendix G.

### 4.4 METRICS AND CALIBRATION DIAGNOSTICS

We assess models with standard metrics for accuracy, uncertainty, and calibration: mean squared error (MSE), predictive variance, continuous ranked probability score (CRPS), risk–coverage curves (area under the risk–coverage curve, AURC), and ID→OOD deltas ($\Delta$MSE, $\Delta$Var, $\Delta$CRPS). For calibration we use both rank- and scale-based diagnostics: (i) Spearman correlation $\rho(\text{Var}, \text{MSE})$ between per-point variance and squared error, and (ii) a linear fit $\text{MSE} \approx a + b\,\text{Var}$, with the ideal calibration corresponding to $a \approx 0$ and $b \approx 1$. We visualize these diagnostics via MSE–Var scatter plots in the ID and OOD regions for toy tasks (Figure 2) and real datasets (Figure 3), together with grouped bar charts of ID→OOD deltas on the real tasks (Figure 4). Precise definitions and Monte Carlo estimators for all metrics are collected in Appendix H.

**Representative seed selection.** To avoid cherry-picking individual runs, all qualitative plots are based on a *representative* seed chosen by a fixed aggregation procedure. For each dataset and configuration we run all models over multiple random seeds (20 seeds per toy task, 100 for Airfoil, and 3 for the remaining UCI datasets) and collect summary metrics per seed (e.g., $\text{MSE}_{\text{ID}}$, $\text{MSE}_{\text{OOD}}$, $\text{Var}_{\text{ID}}$, $\text{Var}_{\text{OOD}}$, $\Delta$CRPS, AURC). For IC-FDNet (or LP-FDNet) we compute the coordinate-wise median of these metrics over seeds and then select the seed whose metric vector is closest (in Euclidean distance) to this median. All per-seed plots in the main text are generated from this representative seed, using the same rule for every dataset, so that the shown behavior is typical rather than hand-picked. On the Airfoil Self-Noise dataset, Figure 5 (Appendix F) visualizes the resulting seed-aggregated MSE–variance scatter, confirming that the selected representative seed lies close to the overall across-seed trend.

### 4.5 RESULTS

**Toy tasks.** Across the three toy tasks (Tables 5–7; Figure 2), FDN's core strength is scale calibration under smooth shifts. On the step and quadratic tasks, IC-/LP-FDNet achieve MSE–Var slopes closer to the ideal $b \approx 1$ with strong rank agreement (Spearman $\rho$ close to 1) and large positive $\Delta$Var, so predictive variance increases in lock-step with difficulty. This comes at the cost of higher AURC and $\Delta$CRPS than the sharpest baselines on the step task, while on the quadratic task their AURC and $\Delta$CRPS are broadly comparable. In contrast, several classical baselines that fit ID sharply (e.g., Deep Ensembles, BayesNet) exhibit much steeper MSE–Var fits ($b \gg 1$) and smaller increases in variance ($\Delta$Var), indicating sharper but less conservative uncertainty even when they rank hard points reasonably well.

On the highly oscillatory sine shift, all methods degrade, but the trade-offs differ. FDN preserves excellent ranking (Spearman $\rho$ near 1) and raises variance substantially OOD (large $\Delta$Var), yet its error grows faster than its variance (large $b$, large $\Delta$MSE), yielding worse AURC. Deep ensembles show smaller $\Delta$MSE and hence better AURC, but their ranking can be weaker. Overall, under matched capacity and update budgets, FDN's main advantages are (i) calibrated scaling on smooth or piecewise-smooth shifts, where many baselines remain overconfident, and (ii) consistently high rank correlation across tasks, which makes FDN a strong triage signal even when absolute scale lags on rapidly oscillatory OOD. This highlights a clear avenue for improvement: stronger variance scaling on such shifts (e.g., temperature/flooring on $\sigma_\phi$, richer priors, or layer-wise $\beta$ schedules).

**Real regression benchmarks.** On the real regression tasks (Airfoil, CCPP, Energy), the same patterns largely persist (Figures 3 and 4; Tables 8–10). FDN achieves reasonable in-distribution MSE and typically exhibits large positive $\Delta\mathrm{Var}$, indicating that uncertainty widens under feature-based shift. This comes with somewhat larger $\Delta\mathrm{MSE}$ and $\Delta\mathrm{CRPS}$ than the sharpest baselines on Airfoil, but more moderate values on CCPP and Energy, so calibration degrades gradually rather than catastrophically. On Airfoil, for example, FDN's ID scatter lies close to the ideal MSE=Var line and spreads out smoothly OOD, with strong Spearman correlation; on CCPP and Energy the predictive variances remain shift-aware and provide useful selective-risk behavior, even when the absolute scale is not always better than the strongest baselines. Taken together, the toy and UCI results suggest that input-conditioned weight stochasticity is a viable and modular route to OOD-aware regression: FDN behaves like a drop-in "uncertainty layer" that can be tuned via $\beta$ and hypernetwork capacity to trade off in-distribution sharpness against OOD conservatism.

Additional qualitative diagnostics, including predictive means (Figure 6, 7), aggregated MSE–variance scatters (Figure 8, 9), and risk–coverage curves for all datasets (Figure 10, 11), are provided in (Appendix F).

## 5 LIMITATIONS

FDN's input-conditioned *weight* stochasticity, like similar stochastic layers, can overfit spurious cues if $\beta$ is too small or the prior is too loose; careful KL scheduling and priors are important. LP-FDN samples weights layer-by-layer, adding latency versus a deterministic pass; sampling at test time also incurs a compute/latency trade-off with $K$, although in our experiments we keep $K$ small. Our study focuses on low-dimensional regression with small/medium tabular datasets and relatively shallow architectures; scaling to high-dimensional inputs (e.g., images), deeper backbones, and structured outputs will require additional engineering, such as low-rank or adapter-style Hypernetworks and more aggressive capacity control.

On the calibration side, FDN still under-scales variance on highly oscillatory OOD regimes (the sine task), and nothing in the current design explicitly enforces frequency-aware or spectral robustness. We also restrict attention to homoscedastic scalar Gaussian likelihoods; heteroscedastic and full-covariance heads, as well as structured priors that tie together different layers or groups of weights, are left to future work. Finally, we only consider regression; applying FDN to classification would require discrete predictive mixtures and calibration metrics beyond CRPS (e.g., ECE/Brier), and may interact non-trivially with common tricks such as label smoothing or temperature scaling.

## 6 CONCLUSION

We introduced Functional Distribution Networks (FDN), which amortize input-conditioned *distributions* over weights to produce predictive densities that remain sharp in-distribution yet expand appropriately under shift. Trained with a Monte Carlo objective and a $\beta$-weighted KL to a simple prior, FDN delivers strong *rank* calibration across tasks and near-ideal *scale* calibration on smooth and piecewise-smooth shifts (step, quadratic), as evidenced by Spearman correlation, MSE–Var slope/intercept, $\Delta\mathrm{Var}$, $\Delta\mathrm{CRPS}$, and AURC. Under a fair protocol that matches parameters, updates, and predictive-sample budgets, FDN remains competitive with standard Bayesian, ensemble, dropout, and hypernetwork baselines on both 1D toy functions and real UCI-style regression benchmarks, and provides uncertainty that is practically useful for abstention and risk-aware inference.

Looking forward, we see several promising directions. On the modeling side, stronger variance scaling mechanisms (temperature/floors, layer-wise $\beta$ scheduling, structured priors) and frequency-aware conditioning may close the remaining gap on highly oscillatory OOD tasks. On the systems side, adapter-style deployments that apply FDN only to a small subset of middle or head layers could allow one to inject uncertainty awareness in new or existing large-scale architectures with modest overhead. Finally, extending FDN to classification, sequence models, and structured prediction—as well as integrating it with other uncertainty-aware modules (e.g., Neural Processes or diffusion-style generative priors over weights)—could yield a general toolkit for calibrated, shift-aware deep learning in practical applications.

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

## A   UNIFIED VIEW VIA $q_\phi(\theta \mid x)$

All methods we consider can be written as

$$p(y \mid x) \;=\; \int p(y \mid x, \theta)\, q_\phi(\theta \mid x)\, d\theta \;\approx\; \frac{1}{K} \sum_{k=1}^{K} p\big(y \mid x, \theta^{(k)}\big), \qquad \theta^{(k)} \sim q_\phi(\theta \mid x).$$

In this paper, architectural layers are specified by the choice of $q_\phi(\theta \mid x)$. Framing models through $q_\phi(\theta \mid x)$ enables apples-to-apples comparisons: (i) how they set the spread of plausible weights, (ii) whether that spread adapts to the input, and (iii) how much compute they expend to form the predictive mixture. FDN's module-level approach directly targets this knob: it provides local, input-aware uncertainty where it is inserted (e.g., the head or later blocks), broadens off-support as inputs drift from the training domain, and leaves the surrounding backbone and training loop unchanged.

**FDN (IC/LP)**   *FDN* makes $q_\phi$ **input-conditional and stochastic**. A common choice is diagonal-Gaussian, factorized by layer:

$$q_\phi(\theta \mid x) = \prod_\ell \mathcal{N}\big(\mu_\ell(x), \operatorname{diag} \sigma_\ell^2(x)\big).$$

This **input-conditional** variant is *IC-FDN*. For **layer-propagated** conditioning (*LP-FDN*), the $\ell$-th layer's weight distribution depends *only* on the previous activation:

$$q_\phi(\theta_\ell \mid x) = \mathcal{N}\big(\mu_\ell(a_{\ell-1}), \operatorname{diag} \sigma_\ell^2(a_{\ell-1})\big), \qquad a_0 := x,$$

and sampling proceeds sequentially across layers along the same Monte Carlo sample path. This induces a first-order Markov structure in depth, allowing uncertainty to expand as signals propagate—later layers can broaden even when early layers remain sharp. We regularize with a per-layer KL:

$$\beta \sum_{\ell=1}^{L} D_{\mathrm{KL}}(q_\phi(\theta_\ell \mid x) \| p_0(\theta_\ell)).$$

More generally, one could condition longer histories $a_{0:\ell-1}$; in this paper, we restrict to first-order (one-step) conditioning. Note, in the limit $\sigma_\ell \to 0$ for all $\ell$, the model collapses to a deterministic layer-conditioned Hypernetwork.

**Deterministic Hypernetwork.**   A deterministic Hypernetwork $G_\phi$ maps the input to weights, yielding a **degenerate** $q$:

$$q_\phi(\theta \mid x) = \delta\big(\theta - G_\phi(x)\big), \qquad p(y \mid x) = p\big(y \mid x, G_\phi(x)\big).$$

Training typically uses NLL or MSE; weight decay on $\phi$ can be interpreted as a MAP prior on the Hypernetwork parameters. Because $q_\phi$ is a Dirac-Delta, there is no weight-space uncertainty: any predictive uncertainty must come from the observation model (e.g., a heteroscedastic head) or post-hoc calibration. Compared to stochastic variants, this adds no KL term and no MC averaging, but can increase per-example compute due to generating weights via $G_\phi$.

**Gaussian HyperNetwork**   A *Stochastic* Hypernetwork outputs a **global** posterior (or context-only):

$$q_\phi(\theta \mid x) \equiv q_\phi(\theta \mid h) = \prod_\ell \mathcal{N}(\mu_\ell(h), \operatorname{diag} \sigma_\ell^2(h)),$$

i.e., independent of the query $x$ (but dependent on a learnable latent task vector $h$). This is variational BNN with parameters produced by a Hypernetwork.

**Bayesian Neural Network (Bayes-by-Backprop).**   A standard variational BNN uses an $x$-**independent** approximate posterior:

$$q_\phi(\theta \mid x) \equiv q_\phi(\theta) = \prod_\ell \mathcal{N}\big(\mu_\ell, \operatorname{diag} \sigma_\ell^2\big),$$

and the same $\beta$-ELBO objective with closed-form diagonal-Gaussian KL. Because $q_\phi(\theta|x)$ is global, predictive uncertainty does not adapt to $x$ except via the likelihood term, which can under-react off-support compared to input-conditional alternatives. On the other hand, the objective is simple and sampling cost is amortized across inputs, though matching ensemble-like diversity typically requires larger posterior variances or multiple posterior samples at test time.

**MLP with Dropout.** MLP with dropout induces a distribution over effective weights via random masks $m$:

$$q_\phi(\theta \mid x) \equiv q_\phi(\theta) \quad \text{(implicit via dropout masks, independent of } x),$$

and inference averages predictions over sampled masks (Gal & Ghahramani, 2016).

**Deep Ensembles.** An $M$-member ensemble of MLPs corresponds to a **finite mixture of deltas**:

$$q(\theta \mid x) \equiv \frac{1}{M} \sum_{m=1}^{M} \delta(\theta - \theta_m), \qquad p(y \mid x) = \frac{1}{M} \sum_{m=1}^{M} p(y \mid x, \theta_m),$$

where each $\theta_m$ is trained independently from a different initialization (and typically a different data order/augmentation). There is no explicit KL regularizer; diversity arises implicitly from independent training trajectories. Inference cost scales linearly with $M$ (one forward pass per member), and for fair comparisons we match total update or compute budgets by reducing epochs.

## B  TRAINING OBJECTIVE

FDN is amortized variational inference with the latent weights $\theta$ and input-conditional posterior $q_\phi(\theta \mid x)$ realized by small hypernetworks via the reparameterization $\theta^{(k)} = g_\phi(x, \varepsilon^{(k)})$ with $\varepsilon^{(k)} \sim \mathcal{N}(0, I)$. For a single $(x, y)$ the ELBO is

$$\log p(y \mid x) \geq \underbrace{\mathbb{E}_{q_\phi(\theta \mid x)}\big[\log p(y \mid x, \theta)\big]}_{\text{data term}} - \underbrace{D_{\mathrm{KL}}\big(q_\phi(\theta \mid x) \,\|\, p_0(\theta)\big)}_{\text{regularizer}}, \tag{1}$$

with a simple prior $p_0(\theta) = \prod_\ell \mathcal{N}(0, \sigma_0^2 I)$.

**(A) $\beta$–ELBO (mean of logs).** We minimize the negative $\beta$–ELBO with $K$ Monte Carlo draws:

$$\mathcal{L}_{\beta\text{-ELBO}} = -\frac{1}{K} \sum_{k=1}^{K} \log p\big(y \mid x, \theta^{(k)}\big) + \beta \, D_{\mathrm{KL}}\big(q_\phi(\theta \mid x) \,\|\, p_0(\theta)\big), \qquad \theta^{(k)} \sim q_\phi(\theta \mid x). \tag{2}$$

Here $\beta{=}1$ recovers standard VI; $\beta \neq 1$ implements capacity control / tempered VI (Higgins et al., 2017; Alemi et al., 2017; Dziugaite & Roy, 2017). We use simple warm-ups for $\beta$ early in training.

**(B) IWAE variant (log of means; tighter bound).** As a reference, the importance-weighted bound is

$$\mathcal{L}_{\mathrm{IWAE}} = -\log\left(\frac{1}{K} \sum_{k=1}^{K} \frac{p_0(\theta^{(k)}) \, p(y \mid x, \theta^{(k)})}{q_\phi(\theta^{(k)} \mid x)}\right), \qquad \theta^{(k)} \sim q_\phi(\theta \mid x), \tag{3}$$

which implicitly accounts for the KL via the weights and typically needs no extra $\beta$ (Burda et al., 2016). We report main results with (A) for simplicity and stability.

**KL decomposition (IC vs. LP).** For *IC-FDN*, layer posteriors condition directly on $x$, so the KL sums over layers and averages over the minibatch. For *LP-FDN*, layer $\ell$ conditions on a sampled hidden state $a_{\ell-1}^{(k)}(x)$; the KL is therefore averaged over this upstream randomness:

$$D_{\mathrm{KL}}\big(q_{\phi,\ell}(\theta_\ell \mid a_{\ell-1}^{(k)}(x)) \,\|\, p_0\big) \quad \text{with} \quad \mathbb{E}_k[\cdot] \text{ across samples } k.$$

With diagonal Gaussians, each layer's closed-form term is

$$D_{\mathrm{KL}}\big(\mathcal{N}(\mu, \mathrm{diag}\,\sigma^2) \,\|\, \mathcal{N}(0, \sigma_0^2 I)\big) = \tfrac{1}{2} \sum_j \left(\frac{\sigma_j^2 + \mu_j^2}{\sigma_0^2} - 1 - \log \frac{\sigma_j^2}{\sigma_0^2}\right),$$

and we implement the variance floor via $\sigma = \varepsilon + \mathrm{softplus}(\rho)$ (no hard clamp).

*Remark.* Future work should investigate *layer-specific* $\beta$ schedules to control where uncertainty is expressed across depth (e.g., larger $\beta$ in early layers for stability, smaller $\beta$ near the output to permit output-scale variance), with the aim of tightening scale calibration ($b \to 1$, $a \to 0$) and improving AURC/CRPS under oscillatory OOD.

## C  NOTATION AND SYMBOLS

Table 1: Key symbols, parameters, and dimensions used throughout the paper.

| Symbol | Description | Dimension / Type |
|---|---|---|
| $x$ | Input / covariate | $\mathbb{R}^{d_x}$ |
| $y$ | Target / response | $\mathbb{R}^{d_y}$ |
| $\mathcal{D}$ | Training dataset | $\{(x_i, y_i)\}_{i=1}^{N}$ |
| $N$ | Number of training examples | $\mathbb{N}$ |
| $T$ | Number of test examples | $\mathbb{N}$ |
| $d_x$ | Input dimension | scalar |
| $d_y$ | Output dimension | scalar |
| $D_y$ | Gaussian head output dim. | $\frac{d_y(d_y+3)}{2}$ |
| $f_\theta$ | Base network (predictive head) | $f_\theta : \mathbb{R}^{d_x} \to \mathbb{R}^{D_y}$ |
| $\theta$ | All base-network weights (all layers) | $\mathbb{R}^P$ |
| $\phi$ | FDN / variational / hypernetwork params | parameter vector |
| $p(y \mid x, \theta)$ | Likelihood (Gaussian head) | $\mathcal{N}(y; \mu_\theta(x), \Sigma_\theta(x))$ |
| $p_0(\theta)$ | Weight prior | $\mathcal{N}(0, \sigma_0^2 I)$ |
| $q_\phi(\theta \mid x)$ | Input-conditioned weight posterior | $\mathcal{N}(\mu_\phi(x), \mathrm{diag}\,\sigma_\phi^2(x))$ |
| $\beta$ | KL weight in $\beta$–ELBO | scalar $\geq 0$ |
| $\mathcal{L}_{\beta\text{-ELBO}}$ | Training loss (per-example) | scalar |
| $L$ | Number of layers in base net | scalar |
| $d_\ell$ | Width of layer $\ell$ | scalar |
| $s_\ell$ | Conditioning signal for layer $\ell$ | $\mathbb{R}^{d_{s,\ell}}$ |
| $d_{s,\ell}$ | Dimension of $s_\ell$ | $d_x$ (IC) or $d_{\ell-1}$ (LP) |
| $A_\ell$ | Per-layer hypernetwork | $A_\ell : \mathbb{R}^{d_{s,\ell}} \to \mathbb{R}^{P_\ell}$ |
| $P_\ell$ | # Gaussian params for layer $\ell$ | $2(d_{\ell-1}d_\ell + d_\ell)$ |
| $P$ | Total # trainable parameters | $\sum_\ell P_\ell$ (plus head) |
| $h_{\mathrm{hyp}}$ | Hypernetwork hidden width | scalar |
| $K$ | MC samples per input (train/test) | $\mathbb{N}$ |
| $M$ | Ensemble size (DeepEnsembleNet) | $\mathbb{N}$ |
| $\sigma_0$ | Prior std. for weights | scalar $> 0$ |
| $\sigma^2$ | Observation noise variance (homoscedastic) | scalar $> 0$ |
| $\varepsilon$ | Variance floor in $\sigma = \varepsilon + \mathrm{softplus}(\rho)$ | scalar $> 0$ |
| $\widehat{\mu}(x)$ | Predictive mean | $\mathbb{R}^{d_y}$ |
| $\widehat{\mathrm{Var}}[Y \mid x]$ | Predictive variance estimator | scalar (for $d_y{=}1$) |
| MSE | Mean squared error | scalar |
| CRPS | Continuous ranked prob. score | scalar |
| NLL | Neg. log predictive density | scalar |
| $\rho$ | Spearman rank correlation | scalar $\in [-1, 1]$ |
| AURC | Area under risk–coverage curve | scalar |
| $\Delta(\cdot)$ | ID→OOD metric delta | $\mathbb{E}_{\mathrm{OOD}}[\cdot] - \mathbb{E}_{\mathrm{ID}}[\cdot]$ |

# D  HETEROSCEDASTIC LIKELIHOOD AND VARIANCE DECOMPOSITION

We briefly collect the forms of the Gaussian $\beta$–ELBO used in this work and the associated decomposition of predictive variance into epistemic and aleatoric components.

**General Gaussian head.** For a Gaussian predictive head with possibly heteroscedastic, full-covariance noise, the per-example $\beta$–ELBO for datum $(x_i, y_i)$ is

$$\mathcal{L}_{\mathrm{Gauss}}^{(i)} = \frac{1}{2K} \sum_{k=1}^{K} \left[ \left\| y_i - \mu_{\theta_k}(x_i) \right\|_{\left(\Sigma_{\theta_k}(x_i)\right)^{-1}}^2 + \log\det\left(2\pi\,\Sigma_{\theta_k}(x_i)\right) \right] + \beta\, D_{\mathrm{KL}}\big(q_\phi(\theta \mid x_i) \,\|\, p_0(\theta)\big),$$

(4)

where $\|v\|_A^2 := v^\top A v$. In the *homoscedastic* case the covariance is constant across inputs, so a full $\Sigma$ encodes a single, global correlation structure among output dimensions; if $\Sigma$ is diagonal, the data term reduces to a (constant-)weighted least-squares plus a constant log-determinant. In

contrast, *heteroscedastic* models use an input-dependent $\Sigma_\theta(x)$, so the weights (and, for full $\Sigma_\theta(x)$, correlations) vary with $x$ and with the sampled weights $\theta$.

For $d_y = 1$ the covariance reduces to a scalar $\sigma_\theta^2(x)$. We say the noise is *homoscedastic in $x$* if $\sigma_\theta^2(x) \equiv \sigma_\theta^2$ (constant across inputs for a fixed $\theta$), and *heteroscedastic in $x$* if $\sigma_\theta^2(x)$ varies with $x$. Orthogonally, because we draw stochastic weights $\theta^{(k)}$, one can distinguish dependence on the sampled weights: *homoscedastic in $\theta$* means $\sigma_\theta^2(x)$ is effectively deterministic (identical across $\theta^{(k)}$ for a given $x$), while *heteroscedastic in $\theta$* means $\sigma_{\theta^{(k)}}^2(x)$ changes with the sampled weights (as in FDN/BNN where the variance head depends on $\theta^{(k)}$).

In the **isotropic heteroscedastic** case, $\Sigma_{\theta^{(k)}}(x) = \sigma_{\theta^{(k)}}^2(x)I$, the Gaussian negative log-likelihood (NLL) contribution is

$$\frac{1}{2K} \sum_{k=1}^{K} \left[ \left( \frac{y_i - \mu_{\theta^{(k)}}(x_i)}{\sigma_{\theta^{(k)}}(x_i)} \right)^2 + \log\big(2\pi\,\sigma_{\theta^{(k)}}^2(x_i)\big) \right], \tag{5}$$

i.e., a *per-input, per-sample weighted* MSE plus a variance penalty. If one instead assumes **isotropic homoscedastic** noise ($\sigma^2$ constant), the data term is proportional to the MSE up to an additive constant; in practice the fit–regularization trade-off can be tuned either by setting $\sigma^2$ or, equivalently, by adjusting $\beta$ to re-balance the data term against the KL.

Since our main focus is on uncertainty-aware metrics rather than cross-output correlations, in the experiments we restrict attention to $d_y = 1$ and use the isotropic homoscedastic case, with a single constant variance $\sigma^2$ that we absorb into $\beta$.

**Scalar homoscedastic $\beta$–ELBO.**  Consider the latent-weight model

$$\theta \sim p_0(\theta), \qquad y \mid x, \theta \sim \mathcal{N}\big(f_\theta(x),\, \sigma^2\big),$$

with a variational family $q_\phi(\theta \mid x)$ (IC-/LP-FDN). For one datum $(x_i, y_i)$ the standard ELBO is

$$\log p(y_i \mid x_i) \geq \underbrace{\mathbb{E}_{q_\phi}\big[\log p(y_i \mid x_i, \theta)\big]}_{\text{data term}} - \underbrace{D_{\mathrm{KL}}\big(q_\phi(\theta \mid x_i) \,\|\, p_0(\theta)\big)}_{\text{regularizer}}.$$

Using the Gaussian likelihood,

$$\log p(y_i \mid x_i, \theta) = -\frac{1}{2\sigma^2}\big(y_i - f_\theta(x_i)\big)^2 - \tfrac{1}{2}\log(2\pi\sigma^2).$$

Plugging into the bound and negating yields the per-example loss

$$\mathcal{L}_{\mathrm{ELBO}}^{(i)} = \frac{1}{2\sigma^2}\,\mathbb{E}_{q_\phi(\theta \mid x_i)}\big[(y_i - f_\theta(x_i))^2\big] + D_{\mathrm{KL}}\big(q_\phi(\theta \mid x_i) \,\|\, p_0(\theta)\big) + \tfrac{1}{2}\log(2\pi\sigma^2).$$

Using $K$ reparameterized samples $\theta^{(k)} \sim q_\phi(\theta \mid x_i)$ gives the unbiased Monte Carlo estimator

$$\boxed{\;\mathcal{L}_{\mathrm{ELBO}}^{(i)} \approx \frac{1}{2K\sigma^2} \sum_{k=1}^{K} \big(y_i - f_{\theta^{(k)}}(x_i)\big)^2 + D_{\mathrm{KL}}\big(q_\phi(\theta \mid x_i) \,\|\, p_0(\theta)\big) + \tfrac{1}{2}\log(2\pi\sigma^2)\;}$$

Since $\tfrac{1}{2}\log(2\pi\sigma^2)$ does not depend on $\phi$ or $\theta$, it can be dropped during optimization. If $\sigma^2$ is fixed, the data term is just a rescaled MSE, so

$$(2\sigma^2)\,\mathcal{L}_{\mathrm{ELBO}}^{(i)} \doteq \frac{1}{K} \sum_{k=1}^{K} \big(y_i - f_{\theta^{(k)}}(x_i)\big)^2 + \underbrace{(2\sigma^2)}_{\beta}\, D_{\mathrm{KL}}\big(q_\phi(\theta \mid x_i) \,\|\, p_0(\theta)\big),$$

showing that choosing constant $\sigma^2$ is equivalent to training with a $\beta$–ELBO, with $\beta$ simply rescaling the effective KL weight (capacity control). In this paper we directly use a $\beta$–ELBO for training.

**Heteroscedastic observation model (scalar case).**    If we allow the observation variance to depend on $x$ and the sampled weights $\theta$,

$$Y \mid x, \theta \sim \mathcal{N}\big(f_\theta(x),\, \sigma_\theta^2(x)\big) \qquad (d_y = 1),$$

the per-example $\beta$–ELBO becomes

$$\mathcal{L}_{\text{het}}^{(i)} = \frac{1}{2K} \sum_{k=1}^{K} \left[ \frac{\big(y_i - f_{\theta^{(k)}}(x_i)\big)^2}{\sigma_{\theta^{(k)}}^2(x_i)} \,+\, \log\big(2\pi\,\sigma_{\theta^{(k)}}^2(x_i)\big) \right] \,+\, \beta\, D_{\text{KL}}\big(q_\phi(\theta \mid x_i) \,\|\, p_0(\theta)\big),$$

i.e., a weighted least-squares (WLS) term plus a variance penalty, with weights $w^{(k)}(x_i) = 1/\sigma_{\theta^{(k)}}^2(x_i)$ learned jointly with the mean.

*Parameterization and stability.* We parameterize

$$\sigma_\theta(x) \,=\, \varepsilon + \text{softplus}\big(\rho_\theta(x)\big), \qquad \varepsilon = 10^{-3},$$

which guarantees positivity and avoids numerical collapse. To mitigate variance blow-up in early training, one can (i) apply gentle weight decay on $\rho_\theta$, (ii) clip $s_\theta(x) = \log \sigma_\theta^2(x)$ to a reasonable range, or (iii) use a short $\beta$ warm-up so the likelihood term dominates initially.

**Predictive variance decomposition.**    Let $\theta \sim q_\phi(\theta \mid x)$ and, given $(x, \theta)$,

$$Y \mid x, \theta \,\sim\, \mathcal{N}\big(\mu_\theta(x),\, \sigma_\theta^2(x)\big).$$

*Scalar case.* The predictive (marginal) variance decomposes as

$$\text{Var}[Y \mid x] \,=\, \underbrace{\mathbb{E}_{\theta \sim q_\phi}\big[\sigma_\theta^2(x)\big]}_{\text{aleatoric}} \,+\, \underbrace{\text{Var}_{\theta \sim q_\phi}\big[\mu_\theta(x)\big]}_{\text{epistemic}}. \tag{6}$$

*Proof.* By the law of total expectation, $\mathbb{E}[Y \mid x] = \mathbb{E}_\theta[\mathbb{E}[Y \mid x, \theta]] = \mathbb{E}_\theta[\mu_\theta(x)]$. By the law of total variance,

$$\text{Var}[Y \mid x] = \mathbb{E}_\theta\big[\text{Var}(Y \mid x, \theta)\big] + \text{Var}_\theta\big(\mathbb{E}[Y \mid x, \theta]\big) = \mathbb{E}_\theta\big[\sigma_\theta^2(x)\big] + \text{Var}_\theta\big[\mu_\theta(x)\big]. \quad \square$$

*Vector-output version.* For $Y \in \mathbb{R}^{d_y}$ with $Y \mid x, \theta \sim \mathcal{N}(\mu_\theta(x), \Sigma_\theta(x))$ the predictive covariance is

$$\text{Cov}[Y \mid x] \,=\, \underbrace{\mathbb{E}_\theta\big[\Sigma_\theta(x)\big]}_{\text{aleatoric}} \,+\, \underbrace{\text{Cov}_\theta\big[\mu_\theta(x)\big]}_{\text{epistemic}}, \tag{7}$$

obtained by the matrix form of the law of total variance.

**Monte Carlo estimators.**    With samples $\theta^{(k)} \sim q_\phi(\theta \mid x)$ we estimate the predictive mean and epistemic variance as

$$\hat{\mu}(x) = \frac{1}{K} \sum_{k=1}^{K} \mu_{\theta^{(k)}}(x), \qquad \widehat{\text{Var}}_{\text{epi}}(x) = \frac{1}{K} \sum_{k=1}^{K} \big(\mu_{\theta^{(k)}}(x) - \hat{\mu}(x)\big)^2,$$

and obtain the total predictive variance via the decomposition equation 6,

$$\widehat{\text{Var}}[Y \mid x] = \frac{1}{K} \sum_{k=1}^{K} \sigma_{\theta^{(k)}}^2(x) \,+\, \widehat{\text{Var}}_{\text{epi}}(x).$$

For $d_y > 1$, replace squared deviations by outer products to estimate covariances, in accordance with equation 7.

In our experiments we use the homoscedastic scalar case ($\sigma_\theta^2(x) \equiv \sigma^2$), so the aleatoric variance reduces to a constant and all input-dependent variability in $\text{Var}[Y \mid x]$ comes from the epistemic component $\text{Var}_\theta[\mu_\theta(x)]$; the heteroscedastic extension above is included for completeness.

## E    FDN TRAINING AND PREDICTION ALGORITHM

---

**Algorithm 1** FDN (Unified for IC-/LP-FDN): Training and Prediction

---

1: **Inputs:** dataset $\mathcal{D} = \{(x_i, y_i)\}_{i=1}^N$; base net $f_\theta$ with layers 1:$L$; per-layer samplers $q_\phi^l(\theta_l \,|\, c_l)$
   (diag. Gaussians); prior $p_0(\theta) = \prod_l p_0^l(\theta_l)$; MC $K$; KL schedule $\{\beta_t\}$; variant $v \in \{\text{IC}, \text{LP}\}$.
2: **for** step $t = 1, 2, \ldots$ **do**
3:     Sample minibatch $\mathcal{B}$; set $\Sigma_{\text{NLL}} \leftarrow 0$, $\Sigma_{\text{KL}} \leftarrow 0$
4:     **for** each $(x, y) \in \mathcal{B}$ **do**
5:       **for** $k = 1, \ldots, K$ **do**
6:         $h_0^{(k)} \leftarrow x$
7:         **for** $l = 1, \ldots, L$ **do**
8:           $c_l \leftarrow \begin{cases} x & \text{if } v = \text{IC} \\ h_{l-1}^{(k)} & \text{if } v = \text{LP} \end{cases}$
9:           $\varepsilon_l^{(k)} \sim \mathcal{N}(0, I), \quad \theta_l^{(k)} \leftarrow \mu_\phi^l(c_l) + \sigma_\phi^l(c_l) \odot \varepsilon_l^{(k)}$
10:          $h_l^{(k)} \leftarrow \text{layer}_l(h_{l-1}^{(k)}; \theta_l^{(k)})$
11:          $\Sigma_{\text{KL}} \mathrel{+}= D_{\text{KL}}(q_\phi^l(\theta_l \,|\, c_l) \,\|\, p_0^l(\theta_l))$
12:         **end for**
13:         $(\mu^{(k)}, \Sigma^{(k)}) \leftarrow \text{head}(h_L^{(k)})$   $\{\Sigma^{(k)}$ may be fixed (homoscedastic)$\}$
14:         $\ell_k \leftarrow \log \mathcal{N}(y; \mu^{(k)}, \Sigma^{(k)})$
15:       **end for**
16:       $\Sigma_{\text{NLL}} \mathrel{+}= -\frac{1}{K} \sum_{k=1}^K \ell_k$   $\{\text{ELBO (mean-of-logs)}\}$
17:     **end for**
18:     $\mathcal{L} \leftarrow \frac{1}{N}(\Sigma_{\text{NLL}} + \beta_t \, \Sigma_{\text{KL}})$;    update $\phi$ by backprop on $\mathcal{L}$
19: **end for**
20: **Predict at** $x_\star$**:** repeat the per-layer sampling with $c_l = \{x_\star \text{ or } h_{l-1}^{(k)}\}$ per $v$ to obtain $(\mu_\star^{(k)}, \Sigma_\star^{(k)})$,
   and return $\hat{p}(y \,|\, x_\star) \approx \frac{1}{K} \sum_k \mathcal{N}(y; \mu_\star^{(k)}, \Sigma_\star^{(k)})$.

---

Table 2: Monte Carlo and training Hyperparameters (defaults unless noted).

| Component | Symbol | Setting |
|---|---|---|
| MC samples (train) | $K_{\text{train}}$ | 1 |
| MC samples (validate) | $K_{\text{val}}$ | 100 |
| MC samples (test) | $K_{\text{test}}$ | 100 |
| Epochs | $\epsilon$ | 400 |
| Optimizer | — | ADAM |
| Learning rate | $\eta$ | $1 \times 10^{-3}$ |
| Batch size | $B$ | 64 |
| Weight prior std | $\sigma_0$ | 1.0 |
| Variance floor | $\varepsilon$ | $10^{-3}$ in $\sigma = \varepsilon + \text{softplus}(\rho)$ |
| KL schedule | $\beta_t$ | linear |
| Maximum $\beta$ | $\beta_{\max}$ | 1.0 |
| Warm up updates | — | 200 |
| Toy Grid Seed | — | 0 |
| Step Seed | — | 13 |
| Sine Seed | — | 19 |
| Quadratic Seed | — | 6 |
| Airfoil Self-Noise Seed | — | 64 |
| CCPP Power Plant Seed | — | 2 |
| Energy Efficiency Heating Seed | — | 2 |
| Stochastic Checkpoint | — | minimum CRPS in interpolation |
| Deterministic Checkpoint | — | minimum MSE in interpolation |

Table 3: Number of training, validation, and test examples used for the toy 1D regression tasks and the three UCI real datasets.

| Dataset | $N_{\text{train}}$ | $N_{\text{val}}$ | $N_{\text{test}}$ | $N_{\text{total}}$ |
|---|---|---|---|---|
| Toy 1D functions | 1024 | 512 | 2001 | 3537 |
| Airfoil Self-Noise | 597 | 199 | 707 | 1503 |
| CCPP Power Plant | 3444 | 1148 | 4976 | 9568 |
| Energy Efficiency (heating load) | 307 | 102 | 359 | 768 |

## F SUPPLEMENTARY FIGURES AND TABLES

In this appendix we provide additional qualitative diagnostics. Figure 6 and Figure 7 plot predictive means versus input for the toy and real tasks, respectively. Figure 8 and Figure 9 show aggregated MSE–variance scatter plots complementing Figure 2 and Figure 3 in the main text, while Figure 10 and Figure 11 report full risk–coverage curves (AURC) for toy and real datasets, complementing the summary statistics in Table 5, Table 6, Table 7, Table 8, Table 9, and Table 10.

Table 4: Model configurations and compute. Columns: base hidden width $d_{\text{hid}}$; hypernetwork hidden width $d_{\text{hyper}}$; latent dim $d_h$ (for Gaussian Hypernetwork); ensemble size $M$; parameter count $P$ (per model). Use "—" where not applicable.

| Model | $d_{\text{hid}}$ | $d_{\text{hyper}}$ | $d_h$ | $M$ | $P$ |
|---|---|---|---|---|---|
| MLPDropoutNet | 333 | — | — | 1 | 1000 |
| Deep Ensemble | 64 | — | — | 10 | 1000 |
| BayesNet | 166 | — | — | 1 | 998 |
| Gaussian HyperNet | 24 | 5 | 9 | 1 | 994 |
| IC-FDNet | 23 | 6 | — | 1 | 1004 |
| LP-FDNet | 24 | 5 | — | 1 | 1011 |

**Notes.** We can see that the parameter count is roughly equal. In order to keep a fair comparison we scale the number of epochs by the ensemble size so the number of updates is roughly the same.

Table 5: Step function: unified calibration/uncertainty summary. Lower is better for AURC and deltas ($\Delta$=OOD–ID); ideal MSE–Var fit has $a \approx 0$, $b \approx 1$.

| Model | $\rho$ | $b$ | $a$ | AURC ↓ | $\Delta$Var (OOD–ID) ↑ | $\Delta$MSE (OOD–ID) ↓ | $\Delta$CRPS (OOD–ID) ↓ |
|---|---|---|---|---|---|---|---|
| MLPDropoutNet | 0.990 | 1.340 | −0.020 | 1.100 | 2.700 | 3.600 | 0.433 |
| DeepEnsembleNet | 0.986 | 4.390 | −0.080 | 3.100 | 2.600 | 11.300 | 1.579 |
| BayesNet | 0.987 | 71.820 | −3.500 | 8.200 | 0.400 | 29.600 | 4.081 |
| GaussHyperNet | 1.000 | 1.020 | 0.590 | 54.000 | 167.300 | 169.800 | 2.386 |
| IC-FDNet | 0.992 | 2.520 | −475.800 | 373.100 | 1 725.400 | 3 820.500 | 16.230 |
| LP-FDNet | 1.000 | 1.040 | −17.930 | 468.500 | 6 306.700 | 6 529.400 | 8.774 |

Table 6: Sine function: unified calibration/uncertainty summary. Lower is better for AURC and deltas ($\Delta$=OOD–ID); ideal MSE–Var fit has $a \approx 0$, $b \approx 1$.

| Model | $\rho$ | $b$ | $a$ | AURC ↓ | $\Delta$Var (OOD–ID) ↑ | $\Delta$MSE (OOD–ID) ↓ | $\Delta$CRPS (OOD–ID) ↓ |
|---|---|---|---|---|---|---|---|
| MLPDropoutNet | 0.966 | 32.680 | 476.110 | 1 178.400 | 112.600 | 4 209.700 | 50.061 |
| DeepEnsembleNet | 0.632 | 1.160 | 1.160 | 1.500 | 1.000 | 1.100 | −0.179 |
| BayesNet | 0.999 | 1.000 | 1.200 | 19.700 | 55.700 | 55.700 | 1.066 |
| GaussHyperNet | 1.000 | 1.020 | 0.850 | 59.000 | 183.800 | 187.000 | 2.265 |
| IC-FDNet | 0.973 | 1.730 | 1 098.340 | 623.400 | 1 436.300 | 3 705.100 | 20.842 |
| LP-FDNet | 0.999 | 1.050 | 38.140 | 382.900 | 3 170.000 | 3 362.400 | 6.527 |

Table 7: Quadratic function: unified calibration/uncertainty summary. Lower is better for AURC and deltas ($\Delta$=OOD–ID); ideal MSE–Var fit has $a \approx 0$, $b \approx 1$.

| Model | $\rho$ | $b$ | $a$ | AURC ↓ | $\Delta$Var (OOD–ID) ↑ | $\Delta$MSE (OOD–ID) ↓ | $\Delta$CRPS (OOD–ID) ↓ |
|---|---|---|---|---|---|---|---|
| MLPDropoutNet | 0.990 | 3 459.120 | −67.320 | 38.700 | 0.100 | 231.700 | 11.176 |
| DeepEnsembleNet | 0.979 | 811.630 | −68.320 | 56.400 | 0.500 | 306.200 | 13.003 |
| BayesNet | 0.953 | 3 855.460 | −89.490 | 81.300 | 0.100 | 389.100 | 15.136 |
| GaussHyperNet | 0.993 | 2.810 | −112.410 | 142.500 | 253.900 | 603.600 | 9.137 |
| IC-FDNet | 0.988 | 1.420 | 225.670 | 377.800 | 1 743.900 | 2 721.700 | 14.073 |
| LP-FDNet | 0.997 | 1.420 | −107.080 | 308.600 | 2 547.300 | 3 510.600 | 10.325 |

Table 8: Airfoil Self-Noise: unified calibration/uncertainty summary. Lower is better for AURC and deltas ($\Delta$=OOD–ID); ideal MSE–Var fit has $a \approx 0$, $b \approx 1$.

| Model | $\rho$ | $b$ | $a$ | AURC ↓ | $\Delta$Var (OOD–ID) ↑ | $\Delta$MSE (OOD–ID) ↓ | $\Delta$CRPS (OOD–ID) ↓ |
|---|---|---|---|---|---|---|---|
| MLPDropoutNet | 0.601 | 10.140 | 0.210 | 0.400 | 0.100 | 1.100 | 0.535 |
| DeepEnsembleNet | 0.177 | 9.740 | 0.570 | 0.600 | 0.000 | 0.800 | 0.353 |
| BayesNet | 0.330 | 4.490 | 0.400 | 0.700 | 0.200 | 1.100 | 0.335 |
| GaussHyperNet | 0.987 | 1.010 | 0.900 | 10.000 | 18.900 | 19.100 | 0.340 |
| IC-FDNet | 0.976 | 1.400 | −3.400 | 7.600 | 19.400 | 25.300 | 0.493 |
| LP-FDNet | 0.983 | 1.000 | 1.010 | 9.300 | 28.500 | 28.800 | 0.363 |

Table 9: CCPP Power Plant: unified calibration/uncertainty summary. Lower is better for AURC and deltas ($\Delta$=OOD–ID); ideal MSE–Var fit has $a \approx 0$, $b \approx 1$.

| Model | $\rho$ | $b$ | $a$ | AURC $\downarrow$ | $\Delta$Var (OOD–ID) $\uparrow$ | $\Delta$MSE (OOD–ID) $\downarrow$ | $\Delta$CRPS (OOD–ID) $\downarrow$ |
|---|---|---|---|---|---|---|---|
| MLPDropoutNet | 0.417 | 6.070 | 0.083 | 0.142 | 0.023 | 0.156 | 0.095 |
| DeepEnsembleNet | 0.126 | 20.050 | 0.127 | 0.139 | 0.002 | 0.071 | 0.065 |
| BayesNet | 0.470 | 1.750 | 0.099 | 0.201 | 0.065 | 0.132 | 0.045 |
| GaussHyperNet | 0.998 | 1.020 | 0.108 | 4.884 | 7.134 | 7.159 | 0.283 |
| IC-FDNet | 0.970 | 1.200 | −0.303 | 2.731 | 2.916 | 3.401 | 0.211 |
| LP-FDNet | 0.902 | 0.990 | 0.903 | 2.844 | 2.129 | 2.820 | 0.260 |

Table 10: Energy Efficiency: unified calibration/uncertainty summary. Lower is better for AURC and deltas ($\Delta$=OOD–ID); ideal MSE–Var fit has $a \approx 0$, $b \approx 1$.

| Model | $\rho$ | $b$ | $a$ | AURC $\downarrow$ | $\Delta$Var (OOD–ID) $\uparrow$ | $\Delta$MSE (OOD–ID) $\downarrow$ | $\Delta$CRPS (OOD–ID) $\downarrow$ |
|---|---|---|---|---|---|---|---|
| MLPDropoutNet | 0.696 | 4.420 | 0.000 | 0.100 | 0.000 | 0.300 | 0.258 |
| DeepEnsembleNet | 0.426 | 7.580 | 0.010 | 0.100 | 0.000 | 0.000 | 0.051 |
| BayesNet | 0.549 | 4.130 | −0.100 | 0.200 | 0.000 | 0.100 | 0.106 |
| GaussHyperNet | 0.996 | 1.040 | 0.420 | 32.400 | 30.300 | 30.900 | 0.454 |
| IC-FDNet | 0.993 | 1.020 | 0.260 | 14.100 | 6.600 | 6.800 | 0.114 |
| LP-FDNet | 0.989 | 0.970 | 1.450 | 14.100 | 6.800 | 6.200 | 0.005 |

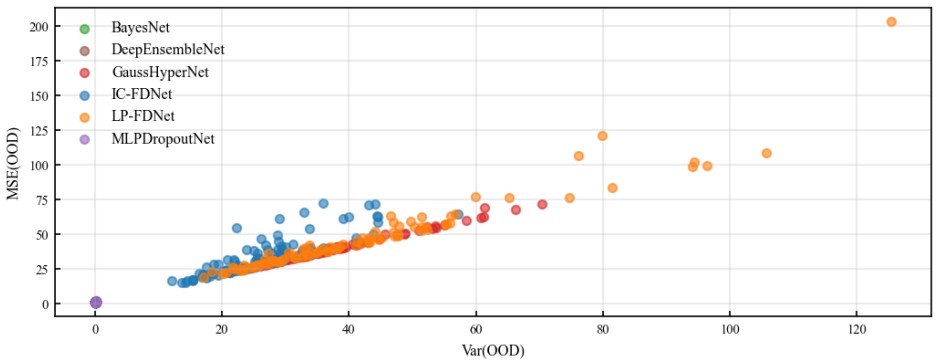

Figure 5: Seed-aggregated MSE–variance scatter on the Airfoil Self-Noise dataset over 100 random initializations. Each point summarizes one seed by its OOD MSE and OOD predictive variance, and the representative seed used in the main-text plots lies close to the overall across-seed trend.

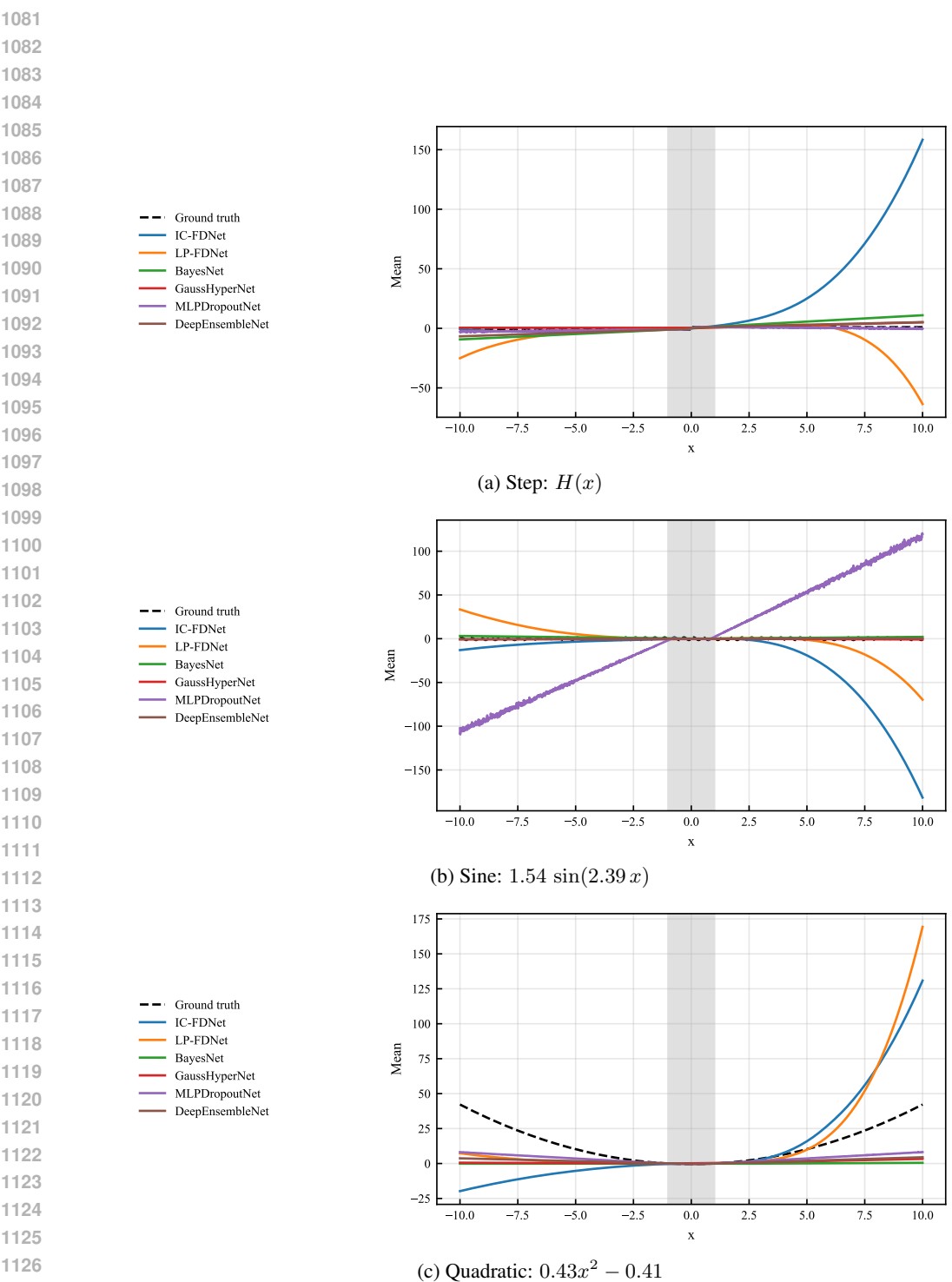

(a) Step: $H(x)$

(b) Sine: $1.54 \sin(2.39\,x)$

(c) Quadratic: $0.43x^2 - 0.41$

Figure 6: Predictive mean vs. input $x$ for the three synthetic 1D toy tasks (step, sine, quadratic), with the ground-truth function overlaid. Shaded region corresponds to the interpolation/ ID points.

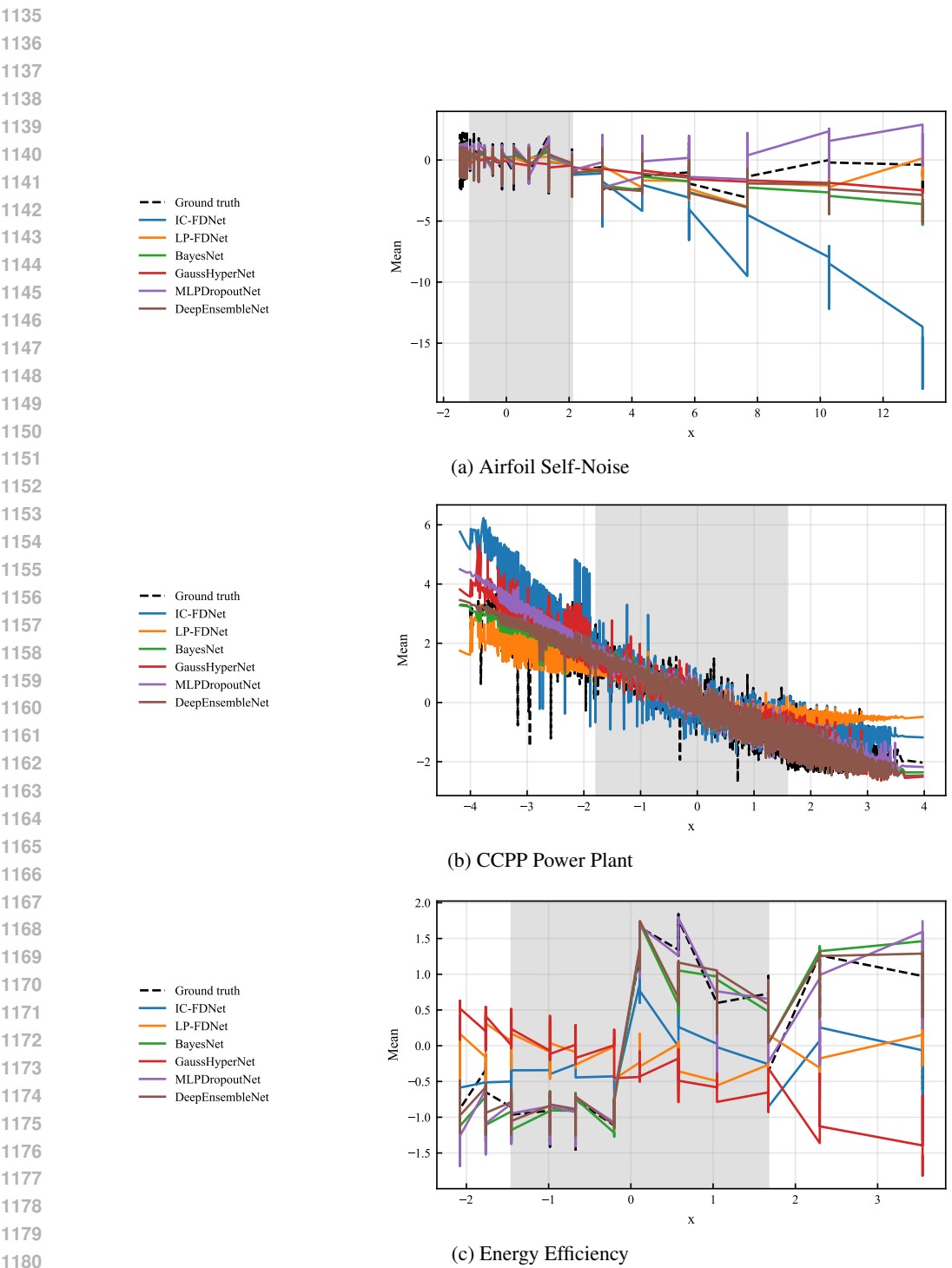

(a) Airfoil Self-Noise

(b) CCPP Power Plant

(c) Energy Efficiency

Figure 7: Predictive mean vs. standardized split feature $x$ for the three real regression datasets (Airfoil Self-Noise, CCPP Power Plant, Energy Efficiency). Shaded region corresponds to the interpolation/ ID points.

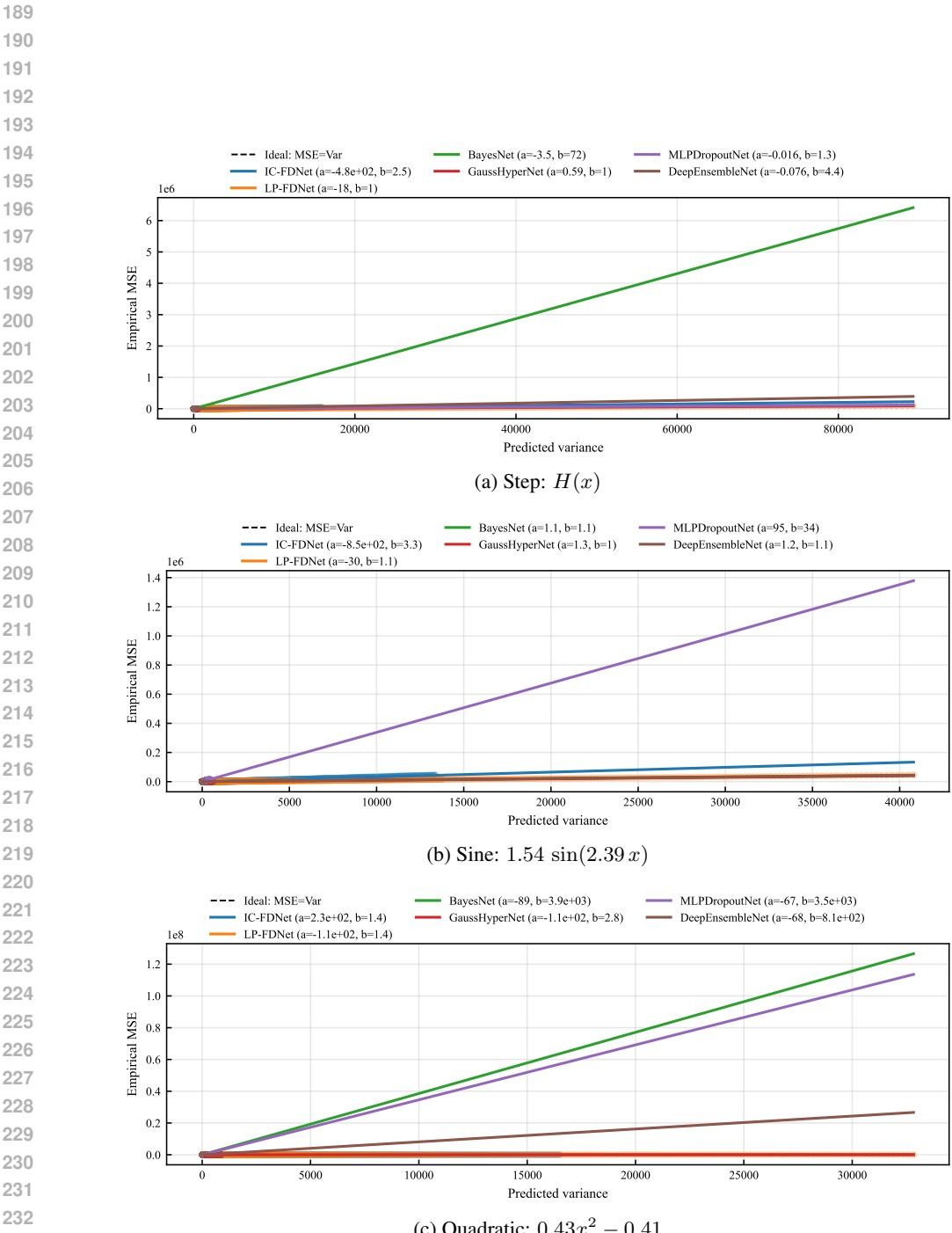

(a) Step: $H(x)$

(b) Sine: $1.54 \sin(2.39\,x)$

(c) Quadratic: $0.43x^2 - 0.41$

Figure 8: MSE vs. predicted variance scatter plots for the three toy tasks. Each panel aggregates ID and OOD test points; the dashed line shows the ideal calibration $\mathrm{MSE} = \mathrm{Var}$. Legends in the PDFs report Spearman's $\rho$ and linear-fit slope/intercept.

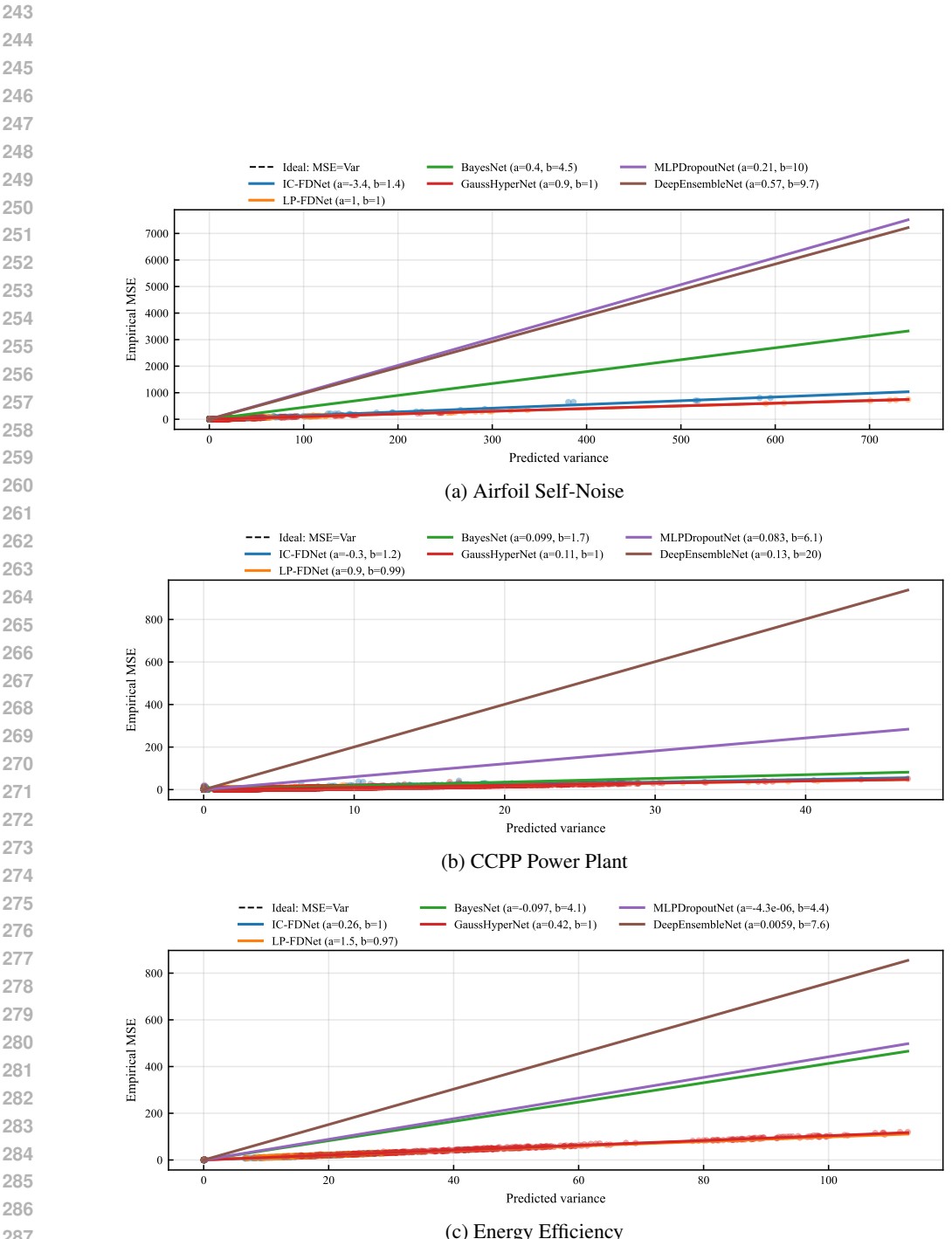

Figure 9: MSE vs. predicted variance scatter plots for the three real datasets. The dashed line marks MSE = Var; legends report Spearman's $\rho$ and linear-fit parameters.

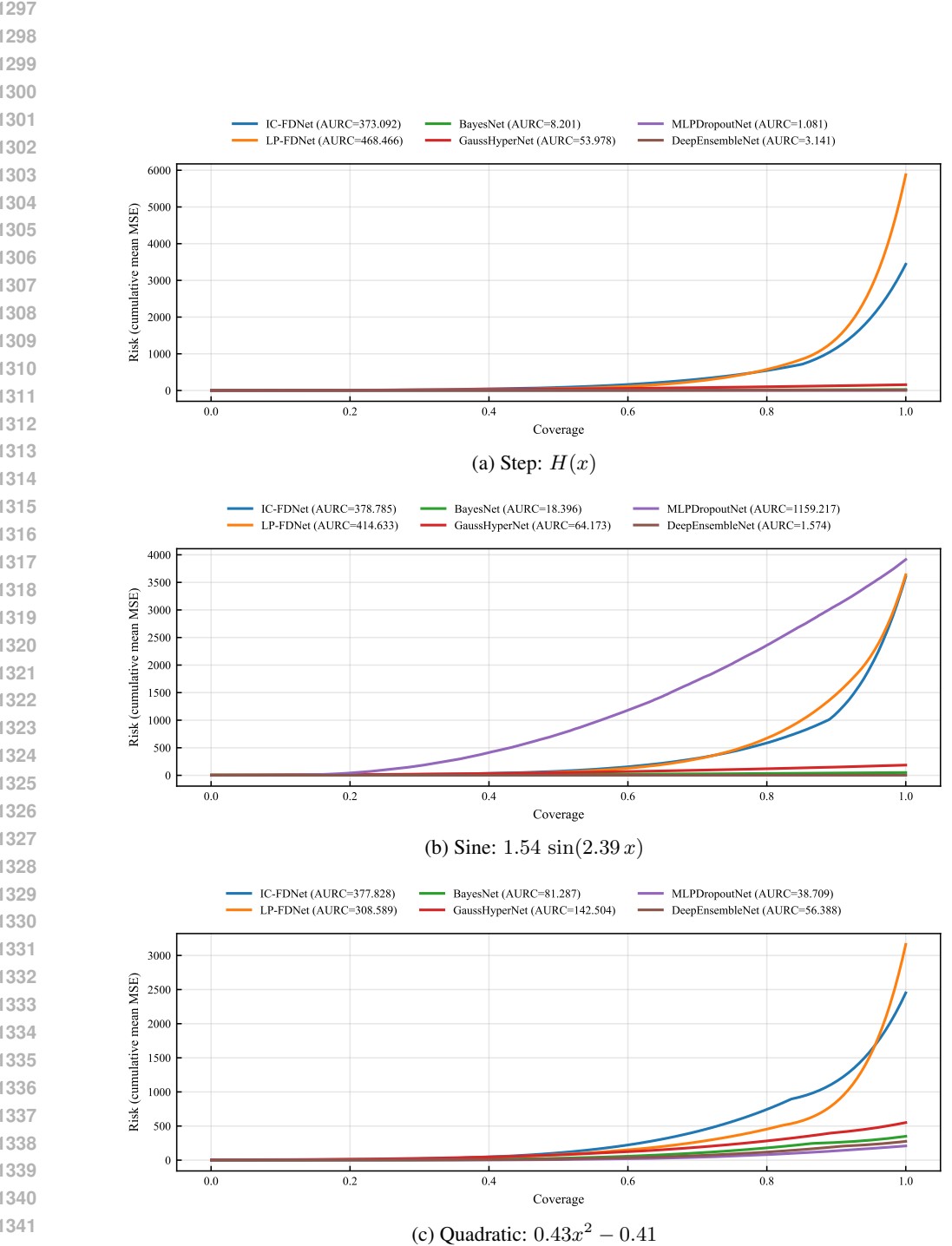

(a) Step: $H(x)$

(b) Sine: $1.54 \sin(2.39\,x)$

(c) Quadratic: $0.43x^2 - 0.41$

Figure 10: Risk–coverage curves (AURC) for the three toy tasks. Curves plot average squared error as a function of coverage as high-variance predictions are rejected. Lower AURC indicates better selective regression.

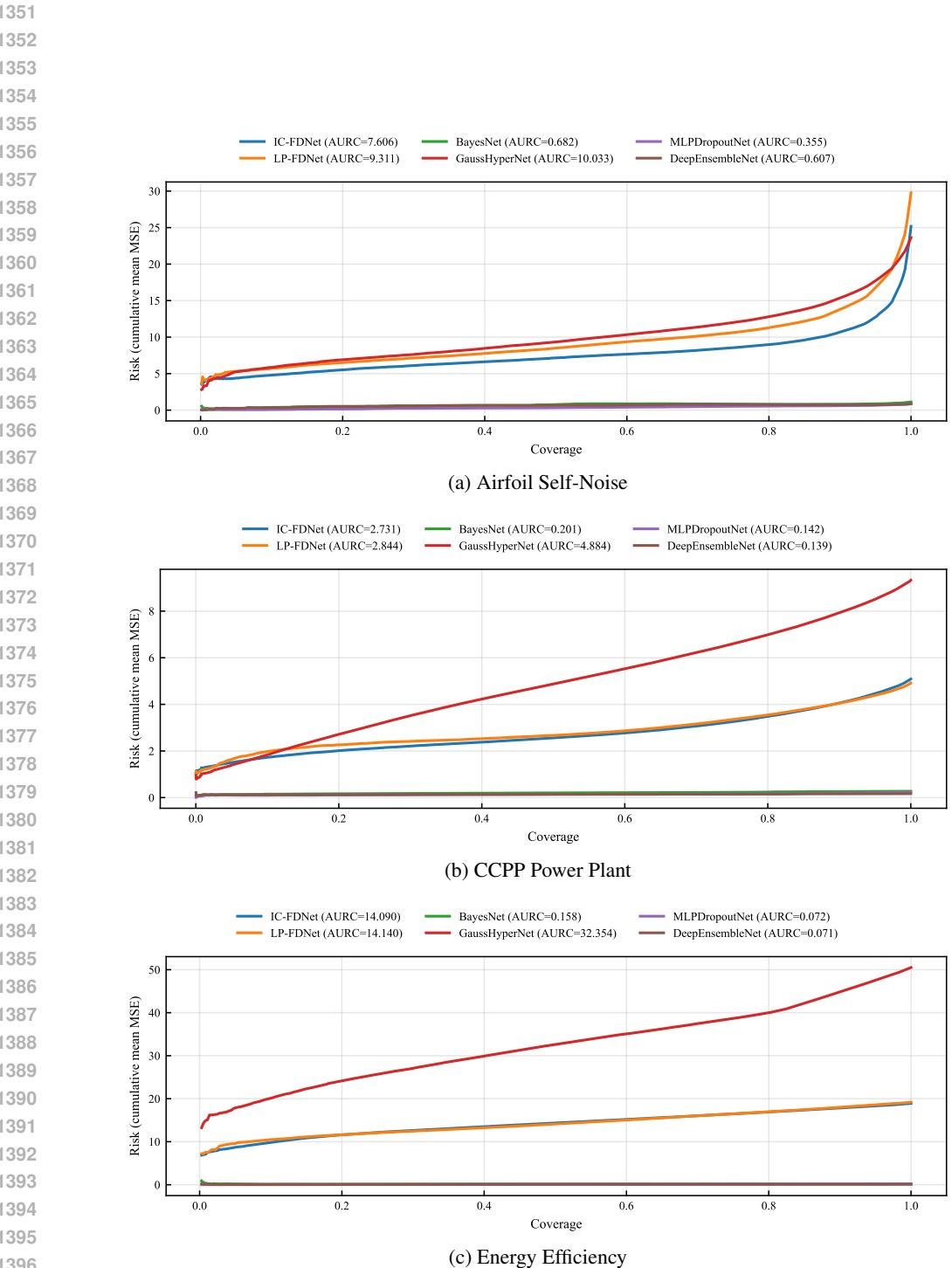

Figure 11: Risk–coverage curves (AURC) for the three real datasets. Lower area under the curve indicates better ability to abstain on high-error points as coverage decreases.

## G   COMPUTATIONAL COMPLEXITY AND SCALABILITY

**Deterministic MLP.**   Let $N$ denote the number of training examples, $d_x$ the input dimension, $d_y$ the output dimension, and consider a base network with widths $\{d_\ell\}_{\ell=0}^L$ with $d_0 = d_x$, $d_L = d_y$. A deterministic MLP has per-epoch cost

$$O\left(N \sum_{\ell=1}^L d_{\ell-1} d_\ell\right),$$

i.e., linear in $N$ and quadratic in the layer widths.

**FDN and Hypernetworks.**   FDN replaces each fixed weight matrix by an input-conditioned diagonal Gaussian $q_\phi(W_\ell, b_\ell \mid s_\ell(x))$ whose parameters are generated by a small per-layer Hypernetwork. For layer $\ell$, the Hypernetwork takes a conditioning vector

$$s_\ell(x) = \begin{cases} x, & \text{IC-FDN}, \\ a_{\ell-1}(x), & \text{LP-FDN}, \end{cases}$$

with dimension $d_{s,\ell} \in \{d_x, d_{\ell-1}\}$, and outputs $(\mu_{W_\ell}, \log \sigma_{W_\ell}^2, \mu_{b_\ell}, \log \sigma_{b_\ell}^2) \in \mathbb{R}^{P_\ell}$, where $P_\ell = 2(d_{\ell-1}d_\ell + d_\ell)$. In our implementation, each Hypernetwork $A_\ell$ is a two-layer MLP with hidden width $h_{\text{hyp}}$. For a mini-batch of size $B$ and $K$ Monte Carlo samples per example, the per-epoch complexity is

$$O\left(NK \sum_{\ell=1}^L \left(d_{s,\ell} h_{\text{hyp}} + h_{\text{hyp}} P_\ell + d_{\ell-1} d_\ell\right)\right).$$

The three bracketed terms correspond to (i) the Hypernetwork input transform, (ii) projection to the $P_\ell$ Gaussian parameters, and (iii) the base-layer matrix–vector product. In all experiments we use $K = 1$ and small Hypernetwork widths (chosen to satisfy a global $\approx 10^3$ parameter budget), making $h_{\text{hyp}} P_\ell$ comparable to $d_{\ell-1} d_\ell$. As a result: (i) the training cost of IC-/LP-FDN remains linear in $N$; (ii) the runtime overhead relative to a deterministic MLP is a small constant factor (typically 2–4×), set by $h_{\text{hyp}}$ and $K$.

**Parameter-count clarification.**   A concern raised in the reviews was that LP-FDN parameter count might scale cubically in model size (e.g., as $O(d_\ell^3)$ in the width of a layer). This would occur only for a specific design where the Hypernetwork directly maps a $d_\ell$-dimensional activation to all $d_\ell^2$ entries of a dense weight matrix via a fully connected layer, which would require a $d_\ell^2 \times d_\ell$ matrix.

In our implementation, however, the Hypernetwork hidden width $h_{\text{hyp}}$ is a small constant, independent of $d_\ell$:

$$P_\ell^{\text{hyper}} = d_{s,\ell} h_{\text{hyp}} + h_{\text{hyp}} \cdot 2(d_{\ell-1}d_\ell + d_\ell) \sim O(h_{\text{hyp}} d_{\ell-1} d_\ell) \sim O(d_{\ell-1} d_\ell).$$

Thus IC-FDN and LP-FDN both have the same $O(d_{\ell-1} d_\ell)$ scaling as the corresponding base MLP layer; in particular, there is no cubic dependence on layer width.

**Architectural trade-offs for larger scales.**   FDN is compatible with standard complexity-reduction techniques without changing the formulation: (i) low-rank factorizations $W_\ell = U_\ell V_\ell^\top$ with the Hypernetwork generating only $(U_\ell, V_\ell)$; (ii) row- or column-wise generation instead of full matrices; and (iii) a shared Hypernetwork with layer-specific output heads. In the main experiments we keep the architecture minimal to match parameter and update budgets across baselines, but these options make FDN readily extendable to higher-dimensional and large-$N$ settings.

**Middle-layer usage and adapter-style deployment.**   In practice, Hypernetworks need not be applied to *every* layer of a deep backbone. FDN is layer-local, so one can restrict stochastic, input-conditioned weights to a small subset of higher layers (or even just the predictive head), keeping earlier blocks deterministic and frozen. This is analogous in spirit to LoRA and adapter modules (Hu et al., 2022; Houlsby et al., 2019): a narrow, trainable "uncertainty adapter" is inserted on top of a largely fixed backbone, so the additional parameters and compute scale with the adapter width rather than with the full network depth or width. In such configurations the overall parameter and FLOP overhead of FDN remains a small fraction of the backbone, even for large architectures, while still enabling input-dependent uncertainty where it is most needed.

## H    EVALUATION METRICS AND CALIBRATION DIAGNOSTICS

For a test set $\{(x_i, y_i)\}_{i=1}^T$ and a stochastic predictor that yields $K$ samples $\{y_i^{(k)}\}_{k=1}^K$ from the predictive distribution $p(y \mid x_i, \mathcal{D})$, we use the following metrics.

**Point prediction error.**    The predictive mean at $x_i$ is

$$\hat{\mu}_i = \frac{1}{K} \sum_{k=1}^K y_i^{(k)}.$$

We define the *per-point* Monte Carlo MSE and bias as

$$\mathrm{MSE}_i = \frac{1}{K} \sum_{k=1}^K \big(y_i^{(k)} - y_i\big)^2, \qquad \mathrm{Bias}_i = \hat{\mu}_i - y_i = \frac{1}{K} \sum_{k=1}^K \big(y_i^{(k)} - y_i\big).$$

For any subset of test inputs $S \subseteq \{1, \ldots, T\}$ (e.g., all, ID, or OOD), we aggregate by averaging over $i \in S$:

$$\mathrm{MSE}_S = \frac{1}{|S|} \sum_{i \in S} \mathrm{MSE}_i, \qquad \mathrm{Bias}_S = \frac{1}{|S|} \sum_{i \in S} \mathrm{Bias}_i.$$

Unless otherwise noted, reported MSE and Bias refer to these region-averaged quantities.

**Predictive variance and epistemic uncertainty.**    The Monte Carlo estimator of predictive variance at $x_i$ is

$$\widehat{\mathrm{Var}}[Y \mid x_i] = \frac{1}{K} \sum_{k=1}^K \big(y_i^{(k)} - \hat{\mu}_i\big)^2,$$

which we aggregate over ID or OOD test splits by averaging across $i$. In the homoscedastic Gaussian setting this decomposes into aleatoric and epistemic components via the law of total variance; we focus on the epistemic part induced by the weight distribution.

**Continuous ranked probability score (CRPS).**    For a univariate predictive CDF $F_i(y)$ and realization $y_i$, the CRPS is

$$\mathrm{CRPS}(F_i, y_i) = \int_{-\infty}^{\infty} \big(F_i(z) - \mathbf{1}\{z \geq y_i\}\big)^2 dz.$$

Using samples $y_i^{(k)} \sim F_i$, we apply the standard Monte Carlo estimator

$$\widehat{\mathrm{CRPS}}_i = \frac{1}{K} \sum_{k=1}^K |y_i^{(k)} - y_i| - \frac{1}{2K^2} \sum_{k=1}^K \sum_{\ell=1}^K |y_i^{(k)} - y_i^{(\ell)}|.$$

We report the average CRPS over ID and OOD test splits. Lower CRPS within a region (ID or OOD) indicates a sharper and better calibrated predictive distribution: mass is concentrated near $y_i$ without being spuriously overconfident. Under shift we typically expect $\mathrm{CRPS}_{\mathrm{OOD}} > \mathrm{CRPS}_{\mathrm{ID}}$ because the task is harder; for a fixed OOD difficulty, smaller $\mathrm{CRPS}_{\mathrm{OOD}}$ (or smaller $\Delta\mathrm{CRPS}$, see below) is better.

**Calibration: MSE–variance relation.**    To assess calibration we compare the predicted variance $\widehat{\mathrm{Var}}[Y \mid x_i]$ with the empirical per-point MSE

$$e_i = \mathrm{MSE}_i = \frac{1}{K} \sum_{k=1}^K \big(y_i^{(k)} - y_i\big)^2.$$

We first compute Spearman rank correlation $\rho = \mathrm{corr}_{\mathrm{Spearman}}(\{\widehat{\mathrm{Var}}_i\}, \{e_i\})$ and fit the linear relation $e_i \approx a + b\,\widehat{\mathrm{Var}}_i$ by least squares; the ideal fit has $a \approx 0$ and $b \approx 1$. High $\rho$ means that larger predicted variance reliably flags larger squared error (good ranking), while $(a, b)$ measure the *scale* of the variances relative to the errors.

In addition, we form *variance–MSE calibration curves* by binning test points into $B$ quantiles of predicted variance. In each bin $b$ (with index set $S_b$) we compute the mean predicted variance $\mathrm{Var}_b = \frac{1}{|S_b|} \sum_{i \in S_b} \widehat{\mathrm{Var}}_i$ and empirical MSE $\mathrm{MSE}_b = \frac{1}{|S_b|} \sum_{i \in S_b} \mathrm{MSE}_i$, and plot the pairs $(\mathrm{Var}_b, \mathrm{MSE}_b)$ together with the ideal $y=x$ line. Points lying close to this diagonal indicate that the typical error magnitude in each confidence bin matches the predicted variance scale.

**Risk–coverage (AURC).** Using variance as an inverse-confidence score, we sort test points by increasing $\widehat{\mathrm{Var}}[Y \mid x_i]$. For a coverage level $c \in (0, 1]$ (fraction of most-confident points retained) we compute the cumulative risk $R(c)$ as the average per-point MSE over the retained subset:

$$R(c) = \frac{1}{|S(c)|} \sum_{i \in S(c)} \mathrm{MSE}_i,$$

where $S(c)$ contains the most-confident fraction $c$ of test points. The area under the risk–coverage curve, $\mathrm{AURC} = \int_0^1 R(c)\,dc$, is estimated numerically. Lower AURC is better: for a fixed difficulty, it means that as we keep only high-confidence predictions, the resulting risk drops more quickly.

**ID vs. OOD deltas.** For each model and dataset we compute MSE, variance, and CRPS separately on ID (interpolation) and OOD (extrapolation) regions, using the region averages defined above, and report

$$\Delta\mathrm{MSE} = \mathrm{MSE}_{\mathrm{OOD}} - \mathrm{MSE}_{\mathrm{ID}}, \quad \Delta\mathrm{Var} = \mathrm{Var}_{\mathrm{OOD}} - \mathrm{Var}_{\mathrm{ID}}, \quad \Delta\mathrm{CRPS} = \mathrm{CRPS}_{\mathrm{OOD}} - \mathrm{CRPS}_{\mathrm{ID}}.$$

For a fixed notion of shift, good uncertainty estimates should exhibit small $\Delta\mathrm{MSE}$ (robust accuracy), *large and positive* $\Delta\mathrm{Var}$ (higher uncertainty OOD than ID), and small $\Delta\mathrm{CRPS}$ (predictive distributions that degrade gracefully rather than collapsing or becoming wildly miscalibrated).

**On NLL / NLPD.** Negative log predictive density (NLL / NLPD) is another strictly proper scoring rule for probabilistic regression and is closely related to CRPS. We computed NLL in preliminary experiments, but found that in our setting it was (i) strongly correlated with CRPS and MSE, and (ii) much more sensitive to occasional extreme errors due to the logarithm, which can dominate the average and obscure more typical behavior. CRPS, by contrast, remains finite, can be estimated directly from samples without specifying a parametric density or bandwidth, and provides a more interpretable summary of the overall predictive distribution (both sharpness and calibration) under dataset shift. For these reasons, and to avoid redundant plots/tables, we report CRPS (together with MSE, variance, and AURC) as our primary proper scoring rule and omit NLL/NLPD from the main results.

LLM DISCLOSURE

ChatGPT assisted with minor copy-editing, LaTeX phrasing, and bibliography chores (suggesting candidate references and drafting BibTeX). The authors independently reviewed the literature and verified all citation metadata (titles, authors, venues, DOIs/arXiv).

