# OpenReview forum: "Functional Distribution Networks (FDN)"
_ICLR.cc/2026/Conference — Submitted to ICLR 2026_

### Official Review · Reviewer_QJsp · 2025-10-30

**Soundness:** 3
**Presentation:** 3
**Contribution:** 3
**Rating:** 6
**Confidence:** 4

**Summary:**

This paper introduces Functional Distribution Networks (FDN), a method that places input-conditioned distributions over network weights to produce predictive mixtures with adaptive dispersion. The approach uses lightweight hypernetworks to amortize a variational posterior $q_{\phi}(\theta|x)$ and trains with a $\beta$-ELBO objective. The authors propose two variants (IC-FDN conditioning only on inputs, and LP-FDN conditioning on previous layer activations) and evaluate them on synthetic regression tasks with an explicit interpolation/extrapolation protocol. Under matched parameter and update budgets, FDN demonstrates competitive in-distribution performance and strong calibration on smooth distribution shifts, though it struggles with highly oscillatory out-of-distribution scenarios.

**Strengths:**

* The paper addresses a critical problem in neural regression (overconfidence under distribution shift) with a principled approach. The explicit evaluation protocol that separates interpolation from extrapolation and uses $\Delta$ Var, MSE-variance slope/intercept, and Spearman correlation provides interpretable diagnostics for uncertainty quality. This makes it clear what "good" OOD behavior should look like (widening uncertainty, positive $\Delta$ Var,, near-unity slope).
* The controlled comparison is commendable, matching parameter budgets (~1000 parameters) and update budgets across all baselines (BNN, Deep Ensembles, Dropout, Hypernetworks). This eliminates common confounders in uncertainty quantification studies where methods differ dramatically in capacity or training cost, making the empirical findings more trustworthy and the conclusions more actionable.
* FDN is positioned as a drop-in replacement for standard MLP layers, making it easy to retrofit into existing architectures without redesigning the predictive head or training pipeline. The use of lightweight hypernetworks to generate layer-wise weight distributions keeps the approach computationally tractable, and the β-ELBO training objective integrates naturally with standard gradient-based optimization.

**Weaknesses:**

* The evaluation is restricted to three synthetic 1D regression tasks, and FDN's calibration breaks down significantly on the sine task (highly oscillatory OOD).  The lack of experiments on higher-dimensional, realistic datasets limits our understanding of how FDN scales beyond toy problems.

* The decision to fix observation variance σ² to a constant is justified as isolating epistemic uncertainty, but it fundamentally limits the model's expressiveness. Real-world regression often exhibits heteroscedastic aleatoric uncertainty that varies with input regions, and forcing homoscedasticity may cause the epistemic component (weight uncertainty) to compensate inappropriately.

* The authors fail to discuss post-hoc calibration techniques for regression models [1,2,3]. I recommend that the authors include a discussion of post-hoc calibration approaches to better ground the paper and provide appropriate context for related work.


**Refs**

[1] Accurate uncertainties for deep learning using calibrated regression

[2] Distribution Calibration for Regression.

[3] Quantile Regularization: Towards Implicit Calibration of Regression Models

**Questions:**

NA

---

> ### Author Response · Authors · 2025-11-24
> **Revision and forthcoming response**
>
> We have uploaded a substantially revised version of the paper. I will post a detailed, point-by-point response to your comments shortly.

---

> ### Author Response · Authors · 2025-11-25
> **QJsp Response**
>
> We thank Reviewer QJsp for the thoughtful and constructive review. We have substantially revised the manuscript to address your main concerns.
>
> (1) Evaluation restricted to synthetic 1D tasks
> The original submission only evaluated on three 1D toy regression tasks. In the revised version we now include three standard UCI-style regression datasets (Airfoil Self-Noise, Combined Cycle Power Plant, and Energy Efficiency) with feature-based ID/OOD splits (Section 4.2). For these real datasets we compute MSE, predictive variance, and CRPS on ID and OOD regions and summarize them via ΔMSE, ΔVar, and ΔCRPS, alongside MSE–Var scatter plots and variance–MSE calibration curves. These experiments show that FDN’s qualitative behavior (shift-aware variance with moderate degradation in error and CRPS) carries over to higher-dimensional, noisy real data, directly addressing your concern about evaluation being limited to three synthetic 1D tasks.
>
> (2) Behavior on the sine/OOD task and highly oscillatory regimes
> You pointed out that FDN underperforms on the sine task in the original submission. In the revised version we make this failure mode explicit. Section 4 and the Limitations section now explain that on the highly oscillatory sine OOD regime, FDN maintains excellent rank calibration (high Spearman ρ) and large ΔVar but its variance under-scales relative to the growth of squared error, leading to worse AURC and CRPS than some baselines. We present this as a genuine limitation of the current architecture and highlight “frequency-aware / spectral robustness” as a concrete direction for future work, rather than glossing over the result.
>
> (3) Fixed homoscedastic Gaussian observation model
> You noted that the main experiments use a scalar homoscedastic Gaussian likelihood and asked about richer predictive heads. In the revision, Section 3.1 clearly states that all main results use a scalar homoscedastic Gaussian head with fixed variance, so that we can focus on epistemic (weight) uncertainty while keeping the predictive head identical across methods for fairness. We have moved the discussion of heteroscedastic and full-covariance Gaussian likelihoods, and more structured priors, to the appendices and explicitly list extending the empirical study to such heads as future work. This makes the scope of the current experiments and the role of the homoscedastic assumption more transparent.
>
> (4) Post-hoc calibration for regression
> You asked about the relation to post-hoc calibration methods for regression. In the revised paper we expand Section 4.4 and the metrics appendix to emphasize CRPS and risk–coverage curves (AURC) as proper, distribution-level calibration diagnostics for regression, playing a role analogous to ECE/AURC in classification. Due to space constraints, we do not include a full empirical comparison to specific post-hoc regression calibration techniques, but FDN is compatible with such methods, and we view “FDN + post-hoc calibration” as a promising direction for future work. If the reviewers consider this comparison essential, we are happy to outline a more detailed discussion in a dedicated appendix.
>
> (5) Code availability
> Finally, we are preparing an anonymized code repository with the full implementation and experiment scripts, which we will share with the reviewers during the discussion phase and make public upon acceptance. We hope these revisions address your concerns and clarify both the strengths and limitations of FDN.

---

> ### Author Response · Authors · 2025-11-28
> **FDN Repo Code Available**
>
> We have released anonymized code and scripts to reproduce all toy and UCI experiments at: https://anonymous.4open.science/r/Functional-Distribution-Networks-6247

---

### Official Review · Reviewer_SLfS · 2025-11-03

**Soundness:** 2
**Presentation:** 2
**Contribution:** 2
**Rating:** 0
**Confidence:** 4

**Summary:**

The paper introduces function distributional network. The main idea is that the neural network weights are sampled from a (variational) distribution that is input dependent, i.e., the parameter distribution can capture input dependent uncertainties. The paper introduces a vartiational bayes method using a beta-VAE to optimize the likelihood. Experimentes are illustrated on very simple 1-D regression tasks.

**Strengths:**

- the paper is relatively well written
- the idea is novel even though I am very sceptical that the approach can be scaled to anything more complex than the presented 1D tasks.

**Weaknesses:**

- the experiments are way too simple, only presenting tiny networks for 1-D regression. the 1-D regression tasks are not even randomized but presented for specific functions, so it is hard to say whether the results are just artifacts of the 3 functions that have been presented. The authors should look into the neural process literature on what family of functions are used there. If we would have the average performance over many instances of the same family, than it would at least be statistically more convincing.
- I do not see how this approach should be scalable in any ways. A standard DNN has O(N^2) parameters (N being the number of neurons per layer). Yet, this approach (at least the LP-FDN) would scale with O(N^3), which is completely unrealistic to use for a reasonable network size. Also the IC-FDN scales poorly with O(D*N^2), where D is the number of input dimensions.
- The plots are not readable (too small font, huge legends etc...) and very hard to understand.

**Questions:**

Please investigate the scaling properties further and present more standardized medium size regression tasks (as promised in the first sentence of the experiment section... but I could not find any other regression tasks except for the 3 almost trivial once presented).

---

> ### Author Response · Authors · 2025-11-24
> **Revision and forthcoming response**
>
> We have uploaded a substantially revised version of the paper. I will post a detailed, point-by-point response to your comments shortly.

---

> ### Author Response · Authors · 2025-11-26
> **UCI Data Sets and Scaling Rebuttal**
>
> We thank Reviewer SLfS for the candid and detailed review. We have substantially revised the manuscript to address your main concerns.
>
> (1) “Experiments are way too simple” / only tiny 1D tasks
> The original submission only evaluated on three 1D toy regression tasks with a single hidden layer. In the revised version we now include three standard medium-size UCI regression benchmarks—Airfoil Self-Noise, Combined Cycle Power Plant, and Energy Efficiency—with feature-based ID/OOD splits (Sec. 4.2). For these datasets we report ΔMSE, ΔVar, and ΔCRPS, and show MSE–Var scatter and variance–MSE calibration curves, under matched parameter and update budgets. These experiments demonstrate that FDN’s main behavior (shift-aware variance with moderate degradation in error and CRPS) carries over to higher-dimensional, noisy real data, directly addressing your request for more standardized, medium-size regression tasks beyond 1D.
>
> For the toy problems, Sec. 4.1 now defines the function families more clearly (step, sine, quadratic), specifies ID vs OOD input ranges, and uses multiple random seeds per task. We view a full Neural-Process-style averaging over many random function instances as an important extension and now list this explicitly as future work in the Limitations.
>
> (2) Scalability and layer-width complexity
>
> In your comment, 𝑁 refers to the number of neurons in a layer (its width). In our notation we use d_ℓ for the width of layer ℓ and reserve 𝑁 for the number of training examples. Your concern was that LP-FDN might scale as 𝑂(𝑁^3) and IC-FDN as O(DN^2) with respect to the layer width.
>
> In the revised version we make the layer-wise costs and parameter count explicit. For a base MLP layer with input width d_ℓ−1 and output width 𝑑_ℓ, the standard dense layer has cost O(d_ℓ−1 d_ℓ). The corresponding FDN layer keeps this same matrix–vector structure and adds a small hypernetwork that generates the weights. If we denote the hypernetwork hidden width by ℎ_hyp, then a typical hypernetwork block has parameter count that scales like O(h_hyp d_ℓ−1 d_ℓ) (or a factorized version) to produce the layer’s weights. In our implementation ℎ_hyp is kept small and fixed across experiments (a narrow MLP), so the hypernetwork cost is a constant-factor multiple of the base layer cost rather than introducing any d_ℓ^2 or d_ℓ^3 interaction between neurons. This follows the standard practice in the hypernetwork literature, where the hypernetwork is explicitly designed to be smaller than the main network and to act as a lightweight weight generator rather than a more expensive replacement.
>
> At the training-loop level (where our  𝑁 denotes the number of training examples), the “Complexity, capacity, and fairness” subsection (Sec. 4.3) and Appendix G show that all models, including IC-/LP-FDNet, have per-epoch cost 𝑂(𝑁_train∑_ℓ𝑑ℓ−1 𝑑_ℓ),
> with the hypernetworks contributing only a modest constant-factor overhead. We also clarify that we deliberately focus on low-dimensional regression with relatively shallow networks under tight parameter/update budgets and discuss that scaling to very deep or very wide architectures would likely benefit from additional engineering (e.g., low-rank or adapter-style hypernets), which we view as future work.
>
> (3) Plot readability
> You noted that the plots in the original version were hard to read (small fonts, large legends). We have re-generated all core figures with larger font sizes, simplified legends, and consistent model ordering and coloring (Figs. 2–4). ID and OOD regions are now clearly separated with shared axes, and captions explicitly state when both are shown. In addition, we added appendix figures showing predictive means, binned MSE–Var curves, and risk–coverage curve, as well as multi-seed scatter for Airfoil: for each experiment we ran multiple seeds (100 for Airfoil, 3 for CCPP and Energy, and 20 per toy task), and then selected the seed whose behavior is closest to the median over all runs, so the displayed patterns are representative rather than cherry-picked.
>
> (4) Request to “investigate scaling properties further and present more standardized medium size regression tasks”
> As noted above, Sec. 4.3 explicitly derives the per-epoch complexity of FDN and baselines, and Sec. 4.2 adds the requested medium-size UCI regression tasks. Together, these revisions address your request to investigate scalability and to move beyond the original three 1D toy functions.
>
> We hope these changes alleviate your concerns about experimental simplicity, scalability, and figure readability, and help clarify the intended scope and practicality of FDN.

---

> ### Author Response · Authors · 2025-11-28
> **FDN Repo Code Available**
>
> We have released anonymized code and scripts to reproduce all toy and UCI experiments at: https://anonymous.4open.science/r/Functional-Distribution-Networks-6247

---

### Official Review · Reviewer_DKN4 · 2025-11-03

**Soundness:** 2
**Presentation:** 2
**Contribution:** 2
**Rating:** 2
**Confidence:** 3

**Summary:**

The paper introduces Functional Distribution Networks (FDNs), a method for per-input amortized variational inference in multi-layer perceptrons (MLPs).

The proposed variational posterior factorizes across layers and each factor is conditioned on either the input location (IC-FDN) or the previous layer's activations (LP-FDN). The layer-wise posteriors are set to be diagonal-covariance Gaussians, and the conditioning is realized via layer-specific hyper-networks, mapping the conditioning signal to the Gaussian's parameters. A zero-mean isotropic Gaussian prior is placed on the MLP parameters, allowing for analytic computation of the standard evidence lower bound objective (ELBO). The variational parameters of the hyper-networks are learned by optimizing a variant of the ELBO, where the KL from posterior to prior is scaled by the regularizer, $\beta$.

The authors compare their approach to a number of uncertainty-quantifying regression methods on three 1D regression tasks.

**Strengths:**

- Quantifying regression predictive uncertainty in a calibrated manner is an unsolved research problem. The paper aims to tackle a highly relevant issue, of great interest to the field.
- The split of the input space into ID and OOD is useful for assessing OOD detection capabilities.
- The proposed method is mostly well explained.
- The positioning of this work relative to related literature is made clear and relevant baselines are included.

**Weaknesses:**

- Figures 1 and 2 are quite hard to understand. This is due to font sizes, legend positioning, overlapping lines and (lack of) scaling of axes. Sometimes there are multiple lines of the same color (e.g. 1e) and it is not clear what they represent. It should be stated clearly in the caption that Figure 1 shows results on both ID and OOD input locations.

- The quantities used to compare methods (CRPS, AURS, ...) should be mathematically defined, including how they are computed on finite data, at least in the appendix.

- The evaluation in general is rather slim. For instance, comparing the MSE of different methods would have been interesting. This would allow statements about the trade-off between uncertainty quantification and accuracy.

- Since these are 1D tasks, a simple plot showing the training data and predictive mean and, e.g., two standard deviations would have been relevant. This would help to build intuition for the model's behavior.

- In Section 3, relevant symbols and their dimensionalities should be explicitly defined. This would make the exposition easier to follow.

- Throughout, more references would be helpful, but particularly in Appendix A.

-  The conclusions one can draw from experiments using a single hidden layer MLP and 1D regression tasks are quite limited. What about higher-dimensional inputs and outputs?

- The step function task is not well defined. Using $\theta$ both for this function and the MLP parameters is suboptimal.

- The code should be made available to reviewers, e.g., via https://anonymous.4open.science.

- The introduction could be sharpened. Discussions of training objective and evaluation protocols may be better put in Sections 3 and 4, respectively.

- The treatment of isotropic, diagonal and full covariance in Section 3.1 seems overly elaborate, given that the observation noise is simply fixed to 1. This would largely be better placed in the appendix.

- Shuffling the rows in Tables 3, 4 and 5 seems unnecessary, same with the columns of the individual plots in Figure 3.

**Questions:**

- Can the authors connect their work to [1]?
- How many training examples are used for each task?
- How should we interpret the huge  $\Delta\text{MSE}$ values for the sin task (e.g. Table 4)? Does this just mean the predictions here are extremely poor?
- Are the training data function values noised?

[1] Jafrasteh, B., Villacampa-Calvo, C., & Hernández-Lobato, D. (2021). Input dependent sparse Gaussian processes.

---

> ### Author Response · Authors · 2025-11-24
> **Revision and forthcoming response**
>
> We have uploaded a substantially revised version of the paper. I will post a detailed, point-by-point response to your comments shortly.

---

> ### Author Response · Authors · 2025-11-25
> **Response to Reviewer DKN4**
>
> We thank Reviewer DKN4 for the careful and detailed review. We have substantially revised the manuscript to address your comments point by point.
>
> (1) Figures and readability
> We re-generated all core figures with larger font sizes, simplified legends, and consistent model ordering and coloring. Overlapping lines were reduced, and we avoided using similar colors for different curves. Captions now explicitly indicate when a figure shows both ID and OOD regions, and the MSE–Var plots clearly separate ID and OOD panels with shared axes.
>
> (2) Definitions of CRPS, AURC, and other metrics
> Section 4.4 and Appendix H now provide mathematical definitions of all metrics used: MSE, predictive variance, CRPS, NLL/NLPD, and risk–coverage (AURC). We also describe how these quantities are estimated from finite datasets and Monte Carlo samples. This directly addresses your request that CRPS, AURC, etc. be defined together with their estimators.
>
> (3) “Slim” evaluation and missing MSE comparisons
> Beyond the original 1D toy tasks, we now include three UCI-style regression benchmarks (Airfoil, CCPP, Energy Efficiency) with feature-based ID/OOD splits (Section 4.2). For both toy and UCI tasks we report ΔMSE, ΔVar, and ΔCRPS, and present MSE–Var scatter plots and variance–MSE calibration curves. This makes the accuracy–uncertainty trade-off explicit (rather than focusing only on calibration) and shows that FDN remains broadly competitive with baselines in terms of error while exhibiting desirable ΔVar and calibration behavior.
>
> (4) Simple plots with mean and uncertainty
> In response to your suggestion, we added qualitative plots in the appendix showing the predictive mean, MSE-bin vs. Var-bin scatter, and risk–coverage curves. These plots are intended to give an intuitive view of interpolation and extrapolation behavior. In addition, we include an MSE–Var scatter plot aggregated over 100 random seeds for the Airfoil dataset, so one can see that the reported behavior is representative rather than cherry-picked.
>
> (5) Notation, dimensionalities, and section organization
> We added a symbol and dimension summary table (Appendix C, Table 1) listing the main symbols (such as d_x, d_y, layer widths, and K) and their shapes. Section 3.2 has been streamlined to introduce IC-FDNet and LP-FDNet more clearly, using a compact variational form q_phi(theta | x) with explicit conditioning signals s_l, together with an updated figure that accurately depicts an FDN layer. We also shortened and refocused the introduction, and moved detailed discussion of the beta-ELBO objective, the evaluation protocol, the algorithm table, and the noise head into Appendices B, D, and E. The main text now focuses on the scalar homoscedastic Gaussian likelihood used in our experiments.
>
> (6) Limited 1D/single-layer setting and the step task
> To go beyond the original 1D, single-layer setting, we added the higher-dimensional UCI experiments mentioned above. Section 4.1 now defines the toy function families more clearly, including the step function (location of the discontinuity and the ID/OOD input ranges), and specifies the architecture used. We also clarify that the toy-task training targets are noise-free to isolate epistemic behavior, whereas the three real datasets naturally contain measurement and feature noise. Studying broader function families and deeper networks is now explicitly listed as future work in the Limitations section.
>
> (7) Sample sizes, noise, and behavior on the sine task
> Appendix F/ Table 3 reports the number of training and test points used for each toy task and for each UCI dataset. As noted, the toy tasks use noise-free function values, while the UCI datasets naturally contain measurement and feature noise. The large ΔMSE values on the sine task arise because this is a highly oscillatory OOD regime in which FDN’s variance under-scales relative to the growth of squared error. We now discuss this explicitly in the results and Limitations and present it as a genuine failure mode: FDN remains well rank-calibrated but under-confident in the tails.
>
> (8) Shuffling of rows and columns in tables and plots
> We have standardized the ordering of models across tables and figures (and their legends), so rows and columns are no longer shuffled between displays. This should make cross-comparison much easier.
>
> (9) Relation to input-dependent sparse Gaussian processes
> We added a short discussion clarifying that FDN can be viewed as a weight-space, hypernetwork-based amortized posterior q_phi(theta | x), which differs from function-space input-dependent sparse GPs such as the work you cited. A full derivation and empirical comparison would require more space and is left for future work, but we agree this is an interesting connection.
>
> (10) Code availability
> We are preparing an anonymized code repository with the full implementation and experiment scripts, which we will share with the reviewers during the discussion phase and make public upon acceptance.

---

> ### Author Response · Authors · 2025-11-28
> **FDN Repo Code Available**
>
> We have released anonymized code and scripts to reproduce all toy and UCI experiments at: https://anonymous.4open.science/r/Functional-Distribution-Networks-6247

---

### Official Review · Reviewer_4gzh · 2025-11-06

**Soundness:** 2
**Presentation:** 2
**Contribution:** 3
**Rating:** 4
**Confidence:** 3

**Summary:**

This paper proposes Functional Distribution Networks (FDN), an architecture where the network’s parameter distribution is conditioned on the input. By dynamically adapting weight distributions, FDN aims to better capture epistemic uncertainty, particularly under out-of-distribution (OOD) conditions. The method is positioned as distinct from Neural Processes, Hypernetworks, and standard Bayesian approaches. Experiments cover both in-distribution (ID) and OOD scenarios, with comparisons to several stochastic baselines and analyses under a fixed parameter budget.

**Strengths:**

- The idea of conditioning parameter distributions on inputs is interesting and well-motivated for handling OOD uncertainty.
- The paper provides a clear conceptual positioning relative to related work.
- The evaluation considers both ID/OOD performance and parameter efficiency.

**Weaknesses:**

- The experimental results do not convincingly show clear gains over strong baselines.
- No real-world experiments are presented, limiting practical validation.
- Figures and presentation could be clearer

**Questions:**

While the approach is interesting, this paper requires more convincing experiments to show the merit of the architecture. I will be open to raising my score if the following questions are addressed:
- Evaluation: How does FDN perform compared to existing methods on real datasets? How about with other standard calibration metrics e.g. ECE and NLL?
- Robustness: Since the parameter distribution can shift a lot depending on the input, how robust is FDN to noise in the data?
- Generalization: How does FDN perform in comparison to other baselines in terms of generalization for ID and OOD? A low $\Delta$MSE in Fig. 3 seems to indicate that the method might be generalizing well even under OOD case.

Additional feedbacks:
- Figures are very hard to read/comprehend (especially Fig. 1)
- Provide more detailed explanations of IC/LP-FDN in the method section
- Line 246 repeats Line 204
- Visualization of the interpolation/extrapolation results would be nice to have.

---

> ### Author Response · Authors · 2025-11-24
> **UCI Regression Data Set Experiments and more**
>
> We thank Reviewer 4gzh for the thoughtful and constructive feedback. We have substantially revised the manuscript to address the main concerns:
>
> (1) Real-world experiments and stronger empirical evidence
>
> UCI regression benchmarks. The original submission only evaluated on 1D toy tasks. In the revised version we now include three standard UCI-style regression datasets—Airfoil Self-Noise, Combined Cycle Power Plant (CCPP), and Energy Efficiency—with explicit feature-based ID/OOD splits (Sec. 4.2). For each dataset we compute MSE, predictive variance, and CRPS on ID and OOD regions and summarize them via ΔMSE, ΔVar, and ΔCRPS, as well as ID/OOD MSE–variance scatter and variance–MSE calibration curves obtained by binning points in predicted variance. Across these real datasets, FDN is generally competitive with baselines and typically exhibits large positive ΔVar with moderate ΔMSE/ΔCRPS, i.e., uncertainty widens under shift while performance degrades gracefully rather than catastrophically.
>
> Robustness to noise. While our toy tasks use noise-free ground truth to isolate epistemic behavior, the new UCI experiments inherently involve measurement and feature noise. On Airfoil and CCPP, for example, FDN remains competitive with baselines in terms of ID error while its variance tracks error under feature shifts, suggesting robustness to realistic noise levels. A more systematic noise–sensitivity study (e.g., controlled corruption experiments) is an interesting direction for future work.
>
> (2) Additional calibration metrics (NLL, relationship to ECE)
>
> NLL/NLPD. Section 4.4 and Appendix H now discuss negative log predictive density (NLL / NLPD) as a strictly proper scoring rule alongside CRPS and MSE. We computed NLL in preliminary experiments and found it strongly correlated with CRPS and MSE in our setting, but highly sensitive to rare extreme errors. The revised text explains why we therefore prioritize CRPS and risk–coverage (AURC) in the main analysis, while acknowledging NLL as a complementary metric.
>
> ECE-style measures. For regression, there is no single canonical ECE analogue; our revised metrics section emphasizes that CRPS and risk–coverage curves play a role similar to ECE/AURC in classification, and we explicitly position our diagnostics as scale- and rank-calibration checks.
>
> (3) Robustness / ID vs OOD behavior
>
> The revised paper clarifies our ID/OOD protocol and adds Δ-metrics (ΔMSE, ΔVar, ΔCRPS) plus MSE–variance scatter plots for both toy and UCI datasets (Sec. 4.1–4.2). We also include variance–MSE calibration curves formed by binning test points in predicted variance. These show that FDN’s predictive variance increases under shift, tracks squared error with near-unity slopes on smooth tasks, and remains competitive in overall performance relative to strong baselines, directly addressing the questions about generalization and robustness under OOD.
>
> Sine/OOD failure analysis. We now explicitly analyze the highly oscillatory sine task in the results and in the Limitations section, noting that FDN maintains excellent rank calibration (ρ ≈ 1) and large ΔVar but still under-scales variance relative to error growth, leading to worse AURC. We also note that nothing in the current architecture enforces frequency-aware or spectral robustness and frame this as a concrete target for future work (Sec. 4.5 and Sec. 5).
>
> (4) Method clarity (IC-FDN / LP-FDN)
>
> Section 3.2 has been streamlined to introduce IC-FDNet and LP-FDNet more clearly, with a compact variational form 𝑞_𝜙(𝜃∣𝑥), explicit conditioning signals 𝑠_ℓ, and a single-layer schematic (Fig. 1) that now matches the notation in the text. A new symbol/dimension summary table in Appendix C (Table 1) further clarifies the mapping between symbols and shapes. With regard to your comment about repetition in this section of the original submission, we have substantially rewritten and condensed the relevant paragraphs in Section 3, which should remove the redundancy you pointed out.
>
> (5) Figures and visualization
>
> We re-generated all core figures with larger font sizes, simplified legends, and consistent model ordering/coloring (Figs. 2–4). The MSE–Var plots now clearly separate ID and OOD panels with shared axes, and the captions explicitly state that both regions are shown. In response to your suggestion, we also added appendix figures showing predictive means, seed-aggregated MSE–variance plots, and risk–coverage curves, providing more intuitive visualizations of interpolation/extrapolation behavior beyond the main-text scatter plots and bar charts.
>
> Finally, we plan to further streamline the figures (e.g., axis labels and legends) during the discussion phase, incorporating any additional suggestions from the reviewers. We are also preparing an anonymized code repository with the full implementation and experiment scripts, which we will share with the reviewers shortly (and publicly upon acceptance). We look forward to your feedback on these revisions.

---

> ### Author Response · Authors · 2025-11-28
> **FDN Repo Code Available**
>
> We have released anonymized code and scripts to reproduce all toy and UCI experiments at: https://anonymous.4open.science/r/Functional-Distribution-Networks-6247

---

### Author Response · Authors · 2025-11-28
**FDN Code Repo Available**

We have released anonymized code and scripts to reproduce all toy and UCI experiments at:
https://anonymous.4open.science/r/Functional-Distribution-Networks-6247

---

### Meta-Review · Area_Chair_zGjw · 2026-01-02

**Summary:**

The paper proposes functional distribution networks, a method to capture uncertainty by dynamically adapting weight distributions. All reviewers highlight that the idea itself is novel and well-positioned regarding related works. They generally commend the existing comparisons, yet unanimously point out that there are several avenues for significant improvement regarding the clarity of writing, complexity of experiments and empirical evidence for scalability, and presentation. The AC agrees with these points and unfortunately believes this paper requires a more major revision before being publishable at a top-tier venue.

**Reviewer Concerns:**

All reviewers shared concerns over the readability, clarity of the presentation, and extent of the experiments regarding problem complexity and scalability. There have been efforts to address these issues in parts, e.g. by making some improvements to figures, adding clarifications in the appendix, and a select set of extra experimentation. The AC acknowledges that these revisions and additions have been conducted, but at the same time, does not believe the reviewer's remarks to be fully addressed to the extent that the paper is substantially improved. The AC believes the paper should conduct a more thorough set of revisions based on the detailed reviewer feedback.

**Reviewer Scores:**

With the exception of one reviewer who gave a borderline positive rating, the other three reviewers shared concerns over the paper not yet being ready for publication. They all pointed out that the idea is great, yet there need to be many more improvements regarding the points mentioned in above summary. Their respective scores differed in extent to which these points have been weighed, with the harshest score being a 0 due to lack of scalability and only 1-D experiments existing in the initial manuscript. Due to the incident, reviewers unfortunately did not have the chance to discuss the rebuttals with the authors. However, the extent of revisions seems unlikely to have potentially swayed all three reviewers to jump to a strictly positive rating. The reviewer giving a 0 had major concerns over experimentation and scalability. Whereas the scalability remark was rectified due to a misunderstanding, the added experimentation is unlikely to match the extent of requested empirical support by the reviewer. Although it is naturally hard to extrapolate what reviewers would have done, the AC believes the reviewer may have raised their score to 2 or 4 at best. Overall, the AC remarks that this paper is tough to accept without the possibility of major revisions and discussions. Given that this was not part of the process at this year's ICLR due to unforeseeable circumstances, the paper thus unfortunately needs to get rejected and undergo another reviewing phase.

---

### Decision · Program_Chairs · 2026-01-26

Reject